

# Understanding aerosol composition in an inter-Andean valley impacted by sugarcane intensive agriculture and urban emissions

Lady Mateus-Fontecha[1], Angela Vargas-Burbano[1], Rodrigo Jimenez*[1], Nestor Y. Rojas[1], German Rueda-Saa [2], Dominik van Pinxteren[3], Manuela van Pinxteren[3], Khanneh Wadinga Fomba[3], Hartmut Herrmann[3]

[1] Universidad Nacional de Colombia – Bogota, Department of Chemical and Environmental Engineering, Air Quality Research Group, Bogota, DC 111321, Colombia
[2] Universidad Nacional de Colombia – Palmira, Department of Engineering and Management, Environmental Prospective, Research Group, Palmira, Valle del Cauca 763533, Colombia
[3] Leibniz Institute for Tropospheric Research (TROPOS), Atmospheric Chemistry Department (ACD), Permoserstrasse. 15, 04318, Leipzig, Germany.

*Correspondence to*: Rodrigo Jimenez (rjimenezp@unal.edu.co)

**Abstract.**

Agro-industrial areas are frequently affected by various sources of atmospheric pollutants that negatively impact public health and ecosystems. However, air quality in these areas is infrequently monitored because of their lower population density compared to large cities, especially in developing countries. The Cauca River Valley (CRV) is an agro-industrial region in Southwest Colombia, where a large fraction of the area is devoted to sugarcane and derivatives production. CRV is also affected by road traffic and industrial emissions. This study aims to elucidate the chemical composition of particulate matter fine mode ($PM_{2.5}$) and to identify the main pollutant sources before source attribution. For this, a sampling campaign was carried out at a representative site of the CRV region, where daily-averaged mass concentrations of $PM_{2.5}$ and the concentrations of water-soluble ions, trace metals, organic and elemental carbon, and various fractions of organic compounds (carbohydrates, n-alkanes, and polycyclic aromatic hydrocarbons – PAHs) were measured. Mean $PM_{2.5}$ was $14.38 \pm 4.35$ µg m$^{-3}$, and the most abundant constituent was organic material ($52.99\% \pm 17.79\%$), followed by ammonium sulfate ($16.12\% \pm 3.98\%$), and elemental carbon ($6.95\% \pm 2.52\%$), which indicates secondary aerosol formation and incomplete combustion. Levoglucosan was present in all samples with a mean concentration of ($113.8 \pm 147.2$ ng m$^{-3}$) revealing biomass burning as a persistent source. The diagnostic ratios applied to organic compounds revealed the influence of petrogenic and pyrogenic sources. Principal component analysis identified the influence of traffic-generated road dust, secondary aerosol formation, gasoline and diesel combustion vehicle exhaust, vegetative detritus, and resuspended agriculture soil. However, no single component was dominant nor explained the CRV $PM_{2.5}$ chemical species variance. Many components had equally important roles instead. Likewise, sugarcane pre-harvest burning, a frequent activity in CRV, was not identified as an independent





component. This aerosol and trace gas source contributed to various components and was correlated to the formation of
secondary aerosols.
Keywords: agro-industry; pre-harvest burning; PM$_{2.5}$; chemical speciation; principal component analysis; Northern South
America

## 1. Introduction


Due to their higher population and population density, air quality in urban areas has disproportionately received much more
attention, from policymakers, governments, and researchers, than rural areas. Sometimes, this is grounded on the
misconception that population sparsity implies lower exposure (Majra, 2011). Especially in developing countries, rural areas
are the least monitored despite the widespread use of high emission practices, including the intensive use of
insecticides/pesticides and fire for land and crop management (Aneja et al., 2008, 2009). Sprayed pesticides release volatile
organic compounds (VOC) that can form tropospheric ozone (Majra, 2011) and secondary organic aerosols (SOA), while
biomass burning emits fine particle matter (PM), black carbon (BC) and trace gases (including CO, CO$_2$, SO$_2$, NO$_x$, NH$_3$,
VOC) that also generate O$_3$ and SOA, all of which affect human health and climate (Yadav and Devi, 2019). Additionally,
agricultural activities are a significant source of nitrogen-containing traces gases (NO$_2$, NO, NH$_3$, N$_2$O) that are released into
the atmosphere from fertilizers, livestock waste and farm machinery (Sutton et al., 2011).

Agricultural burning is worldwide used as an agriculture practice for rapidly and inexpensively clearing the land and for
facilitating tillage and harvesting, so these can proceed unimpeded by external factors. This practice not only results in serious
environmental local issues, like the increase of respiratory diseases for the population that is directly exposed, but also it is
one of the main contributors to global atmospheric pollution (Abdurrahman et al., 2020). Biomass burning is common in
tropical areas of Africa, South America, Asia, and Australia. Although widespread, open agricultural fires are typically shorter
and less intense than forest fires, which make difficult their satellite detection and quantification (Pan et al., 2020). This has
further hindered their observation and analysis. The pre-harvest burning of wheat, corn, rice residues, and sugarcane has been
documented in Mexico(Mugica-Alvarez et al., 2015), Colombia, (Romero et al., 2013), Brazil (Lara et al., 2005; Vasconcellos
et al., 2007) and Thailand (Janta et al., 2019). Sugarcane is a crop of global importance (26.8 million hectares in 2019) (FAO,
2020). About 80% of sugar and almost half of bio-ethanol worldwide are produced from sugarcane cultivated in more than 90
countries, most of them in the Global South but also in some developed economies, including Australia and USA. Sugarcane
pre-harvest burning is still a very common practice worldwide. As other open-field biomass burning practices, sugarcane





burning emits aerosols of high toxicity, secondary organic aerosol precursors and short-lived climate pollutants, among many
other atmospheric contaminants.
Studying the airborne particulate matter chemical composition can be instrumental for the identification of pollutant sources,
including agricultural burning, and the estimation of their contribution to the pollution burden. Most of the field measurement
based studies have been conducted in North America, Europe, and Asia (Karagulian et al., 2015). The number of studies in
Latin America and the Caribbean (LAC) is smaller and have focused on the chemical composition of $PM_{10}$ (Pereira et al.,
2019; Vasconcellos et al., 2011), and source apportionment in urban areas of Colombia (Ramírez et al., 2018; Vargas et al.,
2012), Chile (Jorquera and Barraza, 2012, 2013; Villalobos et al., 2015), Costa Rica (Herrera Murillo et al., 2013) and Brazil
(de Andrade et al., 2010). Previous PM chemical characterization studies in areas with pre- and post-harvest sugarcane burning
have been conducted in Brazil (de Andrade et al., 2010; De Assuncao et al., 2014; Lara et al., 2005; Dos Santos et al., 2002;
Souza et al., 2014; Urban et al., 2012, 2016), and México (Mugica-Alvarez et al., 2015; Mugica-Álvarez et al., 2016). Other
studies have investigated the emissions from sugarcane burning in combustion chambers for $PM_{10}$, $PM_{2.5}$, Elemental Carbon
(EC), Organic Carbon (OC) and PAH (Hall et al., 2012; Jenkins et al., 1992; Mugica-Álvarez et al., 2018). Research in
Colombia is scarce (Romero et al., 2013), even though communities, environmental authorities, and the scientific community
have long recognized the public health, environmental, and climate impacts of open sugarcane burning, especially in the Cauca
River Valley (CRV).

CRV is an inter-Andean valley in Southwest Colombia with a flat area of 5287 $km^2$ (248-km long by 22-km mean width), at
a mean altitude of 985 m MSL (Figure 1), bounded by the Colombian Andes Western and Central Cordilleras, and located at
~120 km from and meteorologically influenced by the Pacific Ocean. CRV encompasses the cities of Cali, Colombia's third-
largest city with 2.2 million inhabitants (hab), Yumbo (129 khab), an important industrial hub, and Palmira (313 khab), which
is the centroid of extensive sugarcane plantations. CRV hosts a highly efficient, resource-intensive sugarcane agro-industry,
with one of the highest biomass yields and the highest sugar productivity in the World (~13 ton sugar/ha) (Asocaña, 2018,
2019). The sugarcane agro-industry produced 3.7% of Colombia's agricultural gross domestic product (GDP) and 2.2% of its
industrial GDP (0.6% of the total GDP) in 2019 (Asocaña, 2019). In 2018 the sugarcane harvest was 195,346 ha, of which
25% belong to 15 sugar mills and 75% to private owners. The production rate was 119.61 sugarcane ton/ha in 2018 and the
average size of each crop is 63 ha, to produce powdered sugar and ethanol used as biofuel. A fraction (45%) of sugarcane is
harvested using a mechanical method and the other fraction (55%) with a manual labor method (Asocaña, 2020). In the manual
method, the crops are burned for some minutes to facilitate the process of cane cutters and this manual harvest also is used as
a socioeconomic tool to provide low-skilled employment to the population of the region. About 69,272 ha (~8.3 Mt) of
sugarcane were burnt in 2018, thus contributing to the emissions of particulate matter (PM) and gases (Cardozo-Valencia et
al., 2019). Since 6.1 Mt of sugarcane bagasse are used to generate electricity (1,657 GWh), this adds additional emissions of
organic components in gases and PM (Asocaña, 2020). Additionally, either pre-harvest burned or not, harvested sugarcane is



transported to mills in multi-car trailers towed by diesel-powered crawlers. The crawler fleet is aged and numerous enough,
and with sufficient annual activity, to potentially constitute an independent source with its own emission chemical profile,
similar to other diesel sources, but with its activity tied to sugarcane harvesting.

For this research purposes only, we made a preliminary estimation of the aggregated $PM_{10}$ emissions in CRV by putting
together disparate source data, including the stationary source emission inventories of CRV's six largest cities excluding
Palmira (Cali, Tulua, Cartago, Jamundi, Yumbo and Buga), Cali's and other cities mobile source emission inventories and an
estimation of sugarcane pre-harvest burning emissions (Cardozo-Valencia et al., 2019), (Table S1). Our preliminary estimation
indicates that the manufacturing industry, with annual emissions of 10.5 kton $PM_{10}$, is the largest $PM_{10}$ emitter in CRV. $PM_{10}$
emissions from mobile sources (3.12 kton $PM_{10}$ year$^{-1}$) and open-field sugarcane burning (1.3 kton $PM_{10}$ year$^{-1}$) are a factor
~3 and ~8 smaller, respectively. Nonetheless, it is worth mentioning the following: 1) The available information was
insufficient for a $PM_{2.5}$ emission estimation; 2) No emission data were available on Palmira, the city in which our measurement
site is located; 3) The stationary emission inventory of Yumbo, an industrial hub with the largest industrial activity, is outdated
and very likely overestimated, particularly as a significant fraction of coal-fired boilers there have been retrofitted to natural
gas. This pollutant source multiplicity, disparity, and uncertainty are indicative of the complexity of the $PM_{2.5}$ source
identification, quantification and location tasks.

This research aimed to characterize the chemical composition of $PM_{2.5}$ at a representative location of CRV, including elemental
carbon (EC), primary and secondary organic carbon (OC), ions, trace metals, and specific molecular markers, including
polycyclic aromatic hydrocarbons (PAH), n-alkanes, and carbohydrates, and to understand the relationships among these
components and with emission sources. Diagnostic ratios and principal component analysis were used to identify the most
important $PM_{2.5}$ components and as a tool for preliminary pollutant source identification, including primary and secondary
aerosols generated by or associated with sugarcane pre-harvest burning (PHB). We believe that in the CRV case, this analysis
is needed prior to source apportionment with receptor models for three reasons: 1) This is the first comprehensive investigation
of particulate matter composition in CRV (prior studies included two types of components at most); 2) There are no suitable
chemical profiles for some pollutant sources, particularly sugarcane PHB; 3) Our measurements dataset is just barely large for
profile-free receptor modeling (positive matrix factorization). Our results are particularly relevant for urban communities and
atmospheres impacted by large-scale intensive agriculture and industrial emissions, particularly in developing countries,
especially in Latin America where PM composition information is still sparse.





## 2. Methods

### 2.1. Description of the sampling site

The sampling site was located on the rooftop of an 8-story administrative building at the Palmira Campus of Universidad Nacional de Colombia (3°30'44.26" N; 76°18'27.40" W, 1065 m altitude), about 27 m above the ground. The Campus is located at the west edge of Palmira's urban area (311 khab), and is surrounded by short buildings at the east and extensive sugarcane plantations, several sugar mills, and other industries elsewhere. Palmira is located at ~27 km northeast of Cali (2.2 Mhab) and ~22 km southeast of Yumbo (123 khab), an important industrial hub. The Pacific Ocean coastline is located at ~120 km across the Western Cordillera , as shown in Figure 1. on the Pacific Ocean coast is one most important international trade seaports in Colombia. Most of the freight is transported by diesel-powered trucks. Road traffic is also substantial within CRV, with Bogota and along the Pan-American highway that connects Colombia with other South American countries .

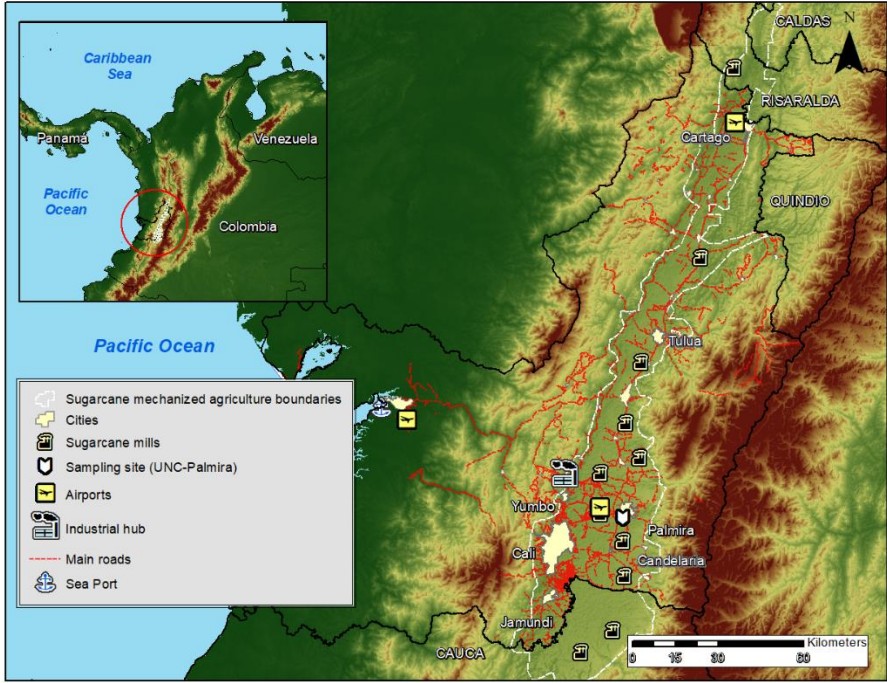

Figure 1. Map of the Cauca River Valley (CRV). The inset shows the location of CRV in Colombia and in Northern South America. The map shows the main cities in CRV, including Palmira (312 thousand inhabitants), our measurement site, and Cali, the most important city in the southwest of Colombia with 2.2 million inhabitants, Yumbo, an industrial hub, and the main highways. Sugar mills, which produce sugar, bio-ethanol, and electric power are also shown. The dash-line delimited area is the CRV's flattest (slope < 5%) bottomland, where mechanized, intensive sugarcane agriculture takes place.





Buenaventura on the Pacific Ocean is one of the busiest ports in Colombia, thus significant diesel combustion emissions occur
along the Buenaventura highway.

**2.2.    Sampling protocols**

The sampling campaign was conducted between 25$^{th}$ July and 19$^{th}$ September 2018. PM$_{2.5}$ aerosol particles (aerodynamic
diameter <2.5 μm) were simultaneously collected on Teflon and quartz fiber filters for 23 h (from 12:00 local time – LT – to
the following day at 11:00 LT), using 2 in-tandem low-volume samplers (ChemComb speciation samplers, R&P). Each
sampler used an independent pump set at a flowrate of 14 L min$^{-1}$. Quartz filters were pre-baked at 600 °C for 8 h before
sampling to eliminate contaminant trace hydrocarbons. In total, 45 samples were collected. Prior to and after exposure, the
filters were conditioned at constant humidity (36±1.5% relative humidity) and temperature (24 ± 1.2 ℃) for 24 h before
weighing them on a microbalance (Sartorius, Mettler Toledo) with 199.99 g capacity and 10 μg resolution. Particulate matter
loaded filters were stored at –20°C until analysis. Mass concentrations were determined from the Teflon filters by differential
weighting. It is worth mentioning that 1888 sugarcane pre-harvest burning events took place during the sampling period. The
vast majority of these events were intentional, controlled, size-limited (~6 ha median area), and short (~25 minutes median
duration) (Fig S1).

**2.3.    Analytical methods**

The quartz-fiber filter samples were analyzed for ions, elemental and organic carbon, and speciation of the carbonaceous
fraction. The Teflon-membrane filter samples were analyzed for metals.

Two circular pieces of quartz filter of 8 mm diameter (100.5 mm$^2$) were punched from the filter and extracted using 1 mL of
ultrapure water (18 MΩ) in a shaker at 400 rpm for 120 min. The extracts were filtered through 0.45 μm syringe filters
(Acrodisc Pall). An aliquot of the solution was analyzed for inorganic (K$^+$, Na$^+$, NH$_4^+$, Mg$^{2+}$, Ca$^{2+}$, Cl$^-$, NO$_3^-$, SO$_4^{2-}$, NO$_2^-$,
PO$_4^{3-}$, Br$^-$, F$^-$) and some organic ions (C$_2$O$_4^{2-}$, CH$_3$O$_3$S$^-$, and CHO$_2^-$) by ion chromatography (IC690 Metrohm; ICS3000,
Dionex). Another aliquot was analyzed for carbohydrates, including levoglucosan, mannosan, and galactosan, as described by
Iinuma et al. (2009a). Organic and elemental carbon was determined from 90.0 mm$^2$ filter pieces following the EUSAAR 2
protocol (Cavalli et al., 2010), with a thermal-optical method using a Sunset Laboratory dual carbonaceous analyzer.

Seventeen metals including K, Ca, Ti, V, Cr, Mn, Fe, Ni, Cu, As, Se, Sr, Ba, Pb, Sn, Sb, and Cu were analyzed from Teflon
and quartz filters by total reflection X-Ray Fluorescence Spectroscopy – TRXF (TXRF, PICOFOX S2, Bruker). Si was not
determined as this element makes part of the quartz filter substrate. Metals were analyzed from three 8-mm circular pieces
punched from the 45 filters, after their digestion with a nitric and chloride acid solution for 180 min to 180 °C. After this, 20





μl aliquots of the digested solution were placed on the surface of polished TXRF quartz substrates along with 10 μl of Ga
solution, which served as internal standard. This solution was left to evaporate at 100°C. The samples were measured at two
angles with a difference of 90° between them to ensure complete excitation of metals. More details on the analytical technique
can be found in Fomba et al. (2013).

Alkanes and polycyclic aromatic hydrocarbons (PAH) were determined from two circular pieces of filter (6 mm diameter, 56.5
mm$^2$), using a Curie-point pyrolyzer (JPS-350, JAI) coupled to a GC-MS system (6890 N GC, 5973inert MSD, Agilent
Technologies). The chemical identification and quantification of the $C_{20}$ to $C_{34}$ n alkanes, along with the following organic
species were performed using the following external standards (Campro, Germany): pristane, phytane, fluorene (FLE),
phenanthrene (PHEN), anthracene (ANT), fluoranthene (FLT), pyrene (PYR), retene (RET), benzo(b)naphtho(1,2-d)thiophene
(BNT(2,1)), cyclopenta(c,d)pyrene (CPY), benz(a)anthracene (BaA), chrysene(+Triphenylene) (CHRY), 2,2-binaphtyl
(BNT(2,2)), benzo(b)fluoranthene (BbF), benzo(k)fluoranthene (BkF), benzo(e)pyrene (BeP), benzo(a)pyrene (BaP), indeno
(1,2,3-c,d)pyrene (IcdP), dibenz(a,h)anthracene (DahA), and benzo(g,h,i)perylene (BghiP), coronene (COR), 9H-Fluoreneone
(FLO(9H)), 9,10-Anthracenedione (ANT (9,10)) and 1,2-Benzanthraquinone (BAQ (1,2)). Four deuterated PAHs,
(acenaphthene-d10, phenanthrene-d10, chrysene-d12, and perylene-d12) and two deuterated alkanes (tetracosane-d50 and
tetratriacontane-d70) were used as internal standards, following the analytical method described by (Neusüss et al., 2000). For
each analyzed compound, sample concentration was calculated by subtracting the average concentration of three blank filters
from the measured concentration.
**2.4.     Mass closure and diagnostic ratios**
PM$_{2.5}$ main components were estimated from the concentrations of EC, OC, water-soluble ions (NO$_3^-$, SO$_4^{2-}$, NH$_4^+$ and Na$^+$)
and tracer metal concentrations (Ca, Ti, Fe, Ni, Cu, Zn, As, Se, Sb, Ba and Pb). The main components considered were organic
material (OM), elemental carbon (EC), ammonium sulfate ((NH$_4$)$_2$SO$_4$), ammonium nitrate (NH$_4$NO$_3$), crustal material (dust),
other trace elements oxides (TEOs), particle-bounded water (PBW), and sea salt (SS), reckoned as sodium chloride. PM$_{2.5}$
closure is described by Eq 1 (Dabek-Zlotorzynska et al., 2011). Except for EC, these components were not directly determined
by chemical analysis but calculated from measured species. For these, we used the Interagency Monitoring of Protected Visual
Environment (IMPROVE) equations (Chow et al., 2015). See Table 1. Also, this reconstruction was instrumental towards the
identification of the main fine airborne particle sources.
The aerosol particle bounded water content was estimated from measured ionic composition, relative humidity, and
temperature following the aerosol inorganic model (AIM) described by (Clegg et al., 1998), available for running online at
http://www.aim.env.uea.ac.uk/aim/model2/model2a.php. AIM describes the thermodynamic equilibrium of the system H$^+$-
NH$_4^+$ - SO$_4^{2-}$ - NO$_3^-$- H$_2$O.






$PM_{2.5}(mass\ closure\ estimated) = OM_{pri} + OM_{sec} + EC + NH_4SO_4 + NH_4NO_3 + Dust + TEO + SS + PBW$     Eq (1)
Table 1. Equations used to estimate the main components of $PM_{2.5}$

| Component | Equation | Reference |
|---|---|---|
| **$OM_{prim}$** | $= f_1\ OC_{prim}$ | (Chow et al., 2015)<br>(Turpin and Lim, 2001) |
| **$OM_{sec}$** | $= f_2\ OC_{sec}$ | (El-Zanan et al., 2005) |
| **$(NH_4)_2SO_4$** | $= 1.3754(SO_4^{2-})_{nss}$<br>Where $(SO_4^{2-})_{nss} = (SO_4^{2-}) - 0.252Na^+$ | (Chow et al., 2015) |
| **$(NH_4)NO_3$** | $= 1.29(NO_3^-)$ | (Chow et al., 2015) |
| **SS** | $= 2.54(Na^+)$ | (Chow et al., 2015)<br>(Snider et al., 2016) |
| **Dust** | $= 1.63Ca + 1.94Ti + 2.42Fe$<br>(Assuming CaO, $Fe_2O_3$, FeO (in equal amounts) and $TiO_2$) | (Chow et al., 2015) |
| **PBW** | $= k\ (SO_4^{2-} + NH_4^+)$ | (Clegg et al., 1998) |
| **TEO** | $= 1.47[V] + 1.27[Ni] + 1.25[Cu] + 1.24[Zn] + 1.32[As] +$<br>$1.2[Se] + 1.07[Ag] + 1.14[Cd] + 1.2[Sb] + 1.12[Ba] +$<br>$1.23[Ce] + 1.08[Pb]$ | (Snider et al., 2016) |

$f_1$ = 1.6. This factor was estimated considering the predominant sources.
$f_2$ = 2.2. This factor was estimated by subtracting the non-carbon component of $PM_{2.5}$ from the measured mass.
$k$ = 0.32 was calculated using the Aerosol Inorganic Model.

The EC tracer method was applied to estimate primary ($OC_{prim}$) and secondary ($OC_{sec}$) organic carbon (Lee et al., 2010). This
method utilizes EC as a tracer for primary OC, which implies that from non-combustible sources $OC_{prim}$ is deemed negligible.
Primary and secondary OC can be estimated upon defining a suitable primary OC to EC ratio ($[OC/EC]_{prim}$). See Eq (2) and
Eq (3). We estimated the $[OC/EC]_{prim}$ ratio as the slope of a Deming linear fit between EC and OC measurements. The term $b$
corresponds to the linear fit intercept, which can be interpreted as the emitted $OC_{prim}$ that is not associated with EC emissions.
This method is limited by the following assumptions: 1) $[OC/EC]_{prim}$ is deemed constant, while in fact this ratio might change
during the day according e.g. to the wind direction and the location of the dominant emission sources. Our 23-h sampling is
expected to smooth this variability source out; 2) It neglects $OC_{prim}$ from non-combustible sources; and 3) Assumes that $OC_{prim}$
is nonvolatile and nonreactive. Departure from these assumptions implies that the estimation of $OC_{prim}$ and $OC_{sec}$ might be
biased, likely underestimating $OC_{sec}$.

222         $OC_{prim} = [OC/EC]_{min} * EC + b$         Eq (2)

223         $OC_{sec} = OC - OC_{prim}$         Eq (3)




As per Table 1, OM was estimated from OC using conversion factors $f_1$ and $f_2$ (Chow et al., 2015), which depend on the OM
oxidation level and the secondary organic aerosol formation and aging during transport. Turpin and Lim, (2001a)
recommended a ratio of 1.6 and 2.1 for urban and non-urban areas, respectively. However, biomass burning aerosols can have
an even higher $f$ values (2.2-2.6), due to the presence of organic components with higher molecular weight, e.g., levoglucosan.
We believe that traffic is the dominant $OC_{prim}$ source at out site, therefore used an $f_1 = 1.6$ to estimate $OM_{pri}$.

We used a factor of 2.2 to estimate $OM_{sec}$ from $OC_{sec}$ fraction. This factor was chosen based on i) recommended ratios of
$2.1\pm0.2$ for aged or non-urban aerosols and ii) the molecular weight to carbon weight ratio for levoglucosan of 2.2.
Levoglucosan is taken as component of reference due to its abundance in samples collected where the biomass burning happens
often and as shown in section 3.6, levoglucosan was a tracer present in whole samples collected in this study (Schauer, 1998).

Concentration ratios among distinct species were used to chemically characterize and infer the main sources of fine particle
matter at Palmira. PM$_{2.5}$ acidity was assessed through cation and anion charge balances and then by comparison of cation
equivalent (CE) and anion equivalent (AE) concentrations (Eq (4) and Eq (5)). Parent PAH ratios are widely used to identify
combustion-derived PAH (Khedidji et al., 2020; Szabó et al., 2015; Tobiszewski and Namieśnik, 2012), although some of
them are photochemically degraded in the atmosphere (Yunker et al., 2002). Additionally, n-alkanes are used as markers of
fossil fuel or vegetation contributions to PM$_{2.5}$. The parameters used to elucidate the n-alkane origin were carbon number
maximum concentration ($C_{max}$), carbon preference index (CPI) and wax n-alkanes percentage (WNA%). Table 2 summarizes
the diagnostic ratio equations and the expected dominant source according to the ratio value.

$$AE = \frac{[SO_4^{2-}]}{48} + \frac{[NO_3^-]}{62} + \frac{[C_2O_4^{2-}]}{44} + \frac{[Cl^-]}{35} + \frac{[PO_4^{3-}]}{31.3} + \frac{[NO_2^-]}{46} + \frac{[Br^-]}{79.9} + \frac{[F^-]}{18.9} + \frac{[CH_3O_3S^-]}{95} + \frac{[CHO_2^-]}{45} \qquad \text{Eq (4)}$$

$$CE = \frac{[Na^+]}{23} + \frac{[K^+]}{39} + \frac{[NH_4^+]}{18} + \frac{[Mg^{2+}]}{12} + \frac{[Ca^{2+}]}{20} \qquad \text{Eq (5)}$$




Table 2. Diagnostic ratios of organic compounds used to infer the sources of PM$_{2.5}$ in this study.

| Diagnostic ratios | Equation | Value | Source | References |
|---|---|---|---|---|
| BeP/(BeP+BaP) | | ~0.5<br>< 0.5 | Fresh particles<br>Photolysis | (Tobiszewski and Namieśnik, 2012) |
| IcdP/(IcdP+BghiP) | | <0.2<br>0.2 - 0.5<br>>0.5 | Petrogenic<br>Petroleum combustion<br>Grass, wood and coal combustion | (Yunker et al., 2002) (Tobiszewski and Namieśnik, 2012) |
| BaP/BghiP | | <0.6<br>>0.6 | Non-traffic emissions<br>Traffic emissions | (Tobiszewski and Namieśnik, 2012) (Szabó et al., 2015) |
| IcdP/BghiP | | >1.25<br><0.4 | Brown coal*<br>Gasoline | (Ravindra et al., 2008) |
| LMW/(MMW+HMW) | | <1<br>>1 | Pyrogenic<br>Petrogenic | (Tobiszewski and Namieśnik, 2012) |
| C$_{max}$ | | < C$_{25}$<br>C$_{27}$ – C$_{34}$ | Anthropogenic<br>Vegetative detritus | (Lin et al., 2010) |
| CPI | $CPI = 0.5 * \left[ \dfrac{\sum_{19}^{33} C_i}{\sum_{20}^{32} C_k} + \dfrac{\sum_{19}^{33} C_i}{\sum_{22}^{34} C_k} \right]$ | CPI ~1<br>CPI > 1 | Fossil carbon<br>Biogenic | (Marzi et al., 1993) (Kang et al., 2018) |
| WNA% | $\sum WNA_{C_n} = [C_n] - \left[ \dfrac{(C_{n+1}) + (C_{n-1})}{2} \right]$<br><br>$WNA\% = \dfrac{\sum WNA_{C_n}}{\sum Total\ n-alkanes}$<br>$PNA\% = 100 - WNA\%$ | WNA ~ 100<br>PNA ~ 100 | Biogenic<br>Anthropogenic | (Lyu et al., 2019) |

249       *Used for residential heating and industrial operation.


As all the measured variables were subject to analytical uncertainty and temporal variability, linear fitting parameters were
obtained from Deming regressions as recommend for atmospheric measurements (Wu and Zhen Yu, 2018). The Spearman
coefficient was selected as an indicator of statistical correlation between chemical components instead of Pearson's to reduce
the effect of outliers. Derived ratios and other parameters were considered statistically significant when p-values < 0.05. The
statistical analysis was made using R version 4.0.2, 24 including the packages corrr (0.4.2), mcr (1.2.1), cluster (2.1.0),
tidyverse (1.3.0), ggplot (3.3.2), psych (2.0.9) and openair (2.7-4).

**2.5. Principal component analysis (PCA)**
There is very little information in the literature on the composition of several of the aerosol emission sources deemed important
in CRV. This is particularly true for sugarcane pre-harvest burning and sugarcane bagasse combustion. Because of this, instead
of directly jumping into a source attribution effort, using receptor modeling methods, we deemed it more important at this





stage of our research to apply multivariate statistical techniques to unravel correlations among the various aerosol components,
and to potentially identify various aerosol sources. For this, we applied principal component analysis (PCA). We consider this
useful in our case, even if PCA is nowadays considered an outdated technique for source attribution in regions with reasonably
characterized sources (Hopke, 2016). The species $Br^-$, $C_{19}H_{40}$, COR, and manosan were excluded from these analyses because
more than 80% of their concentrations were below the detection limit (BDL). Data were organized into a matrix of 45 $PM_{2.5}$
samples (rows) times 73 chemical species (columns). BDL "missing" values were replaced by corresponding species detection
limit. To reduce skewness and order of magnitude effects, the concentration dataset was log10-transformed, mean-centered,
and scaled to unit variance. Principal components were derived from the correlation matrix. We applied varimax rotation PCA
as rotated components have easier-to-interpret loadings. Principal components (PC) were selected to explain at least 60% of
the total variance. Calculations were made with the Psych (2.0.9) R package.

## 3.  Results and discussions

### 3.1.  Meteorology

The Andes Cordillera splits into three south-to-north diverging mountain ranges (Western, Central, and Eastern Cordilleras)
near the Colombia-Ecuador border. The Cauca River Valley (CRV) is an inter-Andean valley at ~985 m altitude located ~120
km from the Pacific Ocean, bounded by the Central and Western Cordilleras (see Figure 1). The Western Cordillera separates
CRV from the Colombian Pacific Ocean watershed, the rainiest region on Earth (Rojo H. and Mesa O., 2020). The elevated
precipitation in this basin is due to the presence of a Walker cell convergence zone at surface, persistent under neutral and La
Niña conditions. This synoptic feature is one the most important determinants of atmospheric circulation in Colombia, with
prevailing east-to-west winds in the lower troposphere along with upper troposphere return winds (Mesa S. and Rojo H., 2020).
The Andean Cordilleras are nevertheless effective barriers to the Walker circulation near the CRV surface (Lopez and Howell,
1967). The elevated humidity in the Pacific Ocean watershed and the closeness of the two Andes branches drive a zonal
regional circulation pattern, consisting in west-to-east anabatic winds over the Pacific slope of the Western Cordillera during
daytime followed by rapid katabatic winds late afternoon (Lopez and Howell, 1967). These winds rapidly ventilate CRV during
the late afternoon – early evening period on an almost regular basis. CRV is wide (~22 km) and long (~248 km) enough to
develop a valley-mountain wind circulation pattern during daytime. Winds are very mild during this time period and expectedly
highly dispersive, i.e. with high turbulence intensities (Ortiz et al., 2019). The arrival of the katabatic "tide" at late afternoon
wipes the valley-mountain wind pattern out.
One year prior to the sampling period, we monitored the local meteorology, first at 14.5 m, a few meters over the mean canopy
level, and then at 32.5 m during the sampling campaign. The box-and-whisker plot in Fig 2 shows katabatic tide winds of up
to ~8 m/s at the sampling site elevation, peaking at ~17:00 LT. Wind speeds were a factor ~2-3 slower at ground level. The
wind runs at the sampling height were typically over ~200 km per day (Fig S3) indicating that the samples had quite a large





spatial coverage of CRV, much larger than it would have been at ground level. This also implies that the samples were
frequently and significantly influenced by emissions coming from Yumbo's industrial hub (northwest of Palmira), and also by
Palmira and Yumbo urban and highway emissions, along with pre-harvest sugarcane burning and sugarcane mill emissions.
The wind rose (Fig 2a) suggests that the influence of urban emissions from Cali, CRV's largest city by far, was minor. Other
meteorological variables are reported in the Supplementary Material (SM) (Fig S2). Temperature (24.2°C on average) and
relative humidity (71.6%) were very likely controlled by solar radiation (350 W m$^{-2}$ on average). The pressure daily profile
(~763 hPa on average) clearly showed the influence of the katabatic tide, with a ~3 hPa drop during its arrival at late afternoon.
Overall, we believe our measurements at the Palmira site are quite representative of the regional air quality.

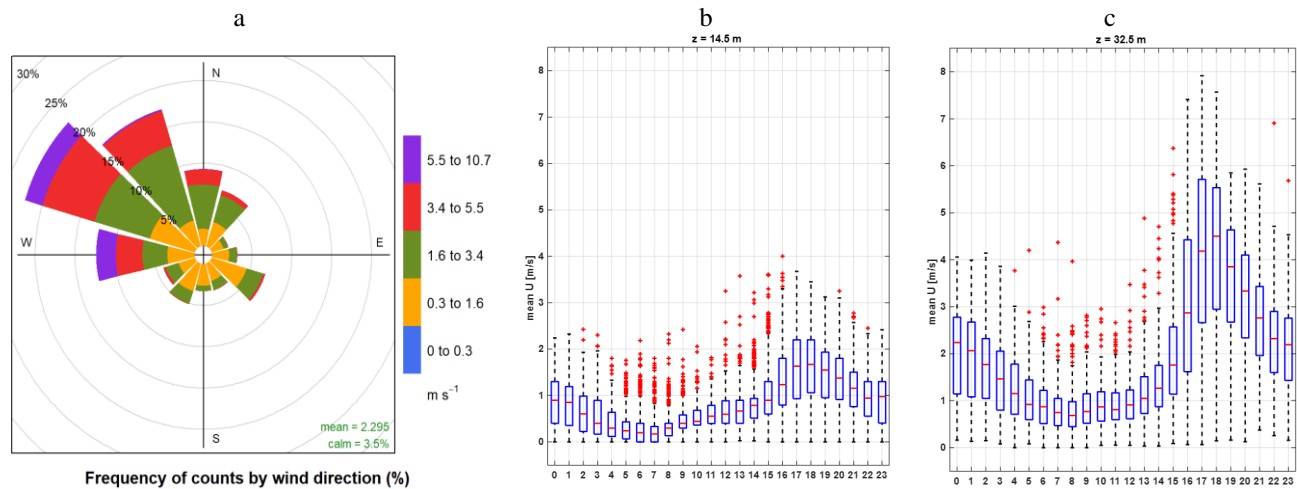

Figure 2. Wind pattern in the sampling location: a) predominant wind rose during sampling period, b) hourly profile of wind
speed to 14.5 m over ground level and c) hourly profile of wind speed in sampling location at 32.5 m over the ground level.

**3.2.   Bulk PM$_{2.5}$ concentration and composition**

The daily PM$_{2.5}$ concentration measured in this study ranged from 6.73 to 24.45 µg m$^{-3}$ with a campaign average of 14.38 ±
4.35 µg m$^{-3}$ (23 h-average, ±1-sigma). Although these concentrations may appear comparatively low, it is worth stressing that
samples were collected at more than 30 m height with hourly wind speeds frequently above 4 m s$^{-1}$.

Previous studies conducted in rural areas of Brazil impacted by open field sugarcane burning reported significantly higher
(mean 22.7 µg m$^{-3}$; Lara et al., 2005), similar (mean 18 µg m$^{-3}$Souza et al., 2014), and significantly lower PM$_{2.5}$ concentrations
(mean 10.88 µg m$^{-3}$; Franzin et al., 2020). Comparable measurements in Mexico during harvest periods showed much higher
concentrations, from 29.14 µg m$^{-3}$ (Mugica-Alvarez et al., 2015) up to 51.3 µg m$^{-3}$ (Mugica-Álvarez et al., 2016). Our PM$_{2.5}$
concentration measurements in CRV are thus substantially lower than those usually reported in Mexico and Brazil during





sugarcane burning periods. Major differences among sugarcane pre-harvest burning practices in Colombia and Brazil and
Mexico must be considered while comparing concentrations. First, currently in CRV, "just" ~1/3 of the sugarcane area is
burned before harvesting compared to much larger fractions in Mexico and Brazil. Second, sugarcane is harvested year-long
in CRV compared to Brazil and Mexico, where harvest is limited to a ~6-month period (known as *zafra* in Spanish, "the
harvest"). Third, the size of the individual plots burned in CRV is typically ~6 ha (median burned area; Cardozo-Valencia et
al., 2019), compared to much larger plots and total areas in Brazil and Mexico.

OC was the most abundant measured component of $PM_{2.5}$ with a mean daily concentration of $3.97 \pm 1.31$ µg m$^{-3}$, whereas the
mean EC concentration was only $0.96 \pm 0.31$ µg m$^{-3}$. These two contributed to $29.1 \pm 8.3\%$ and $7.2 \pm 2.3\%$ of the $PM_{2.5}$ mass,
respectively (carbonaceous fractions were thus $4.93 \pm 1.58$ µg m$^{-3}$, i.e. $36.31 \pm 10.41\%$ of $PM_{2.5}$). The most abundant water-
soluble ions found in Palmira's $PM_{2.5}$ were $SO_4^{2-}$, $NH_4^+$, and $NO_3$, with average concentrations of $2.15 \pm 1.39$ µg m$^{-3}$, $0.67 \pm$
$0.62$ µg m$^{-3}$, and $0.51 \pm 0.30$ µg m$^{-3}$, respectively ($12.7 \pm 2.8\%$, $3.7 \pm 1.1\%$ and $2.6 \pm 1.3\%$ of mass concentration, respectively).
Mean concentrations of other water-soluble ions, such as $Na^+$, $Ca^+$, and $C_2O_4^{2-}$, were around 0.1 µg m$^{-3}$, while those of $K^+$,
$PO_4^{3-}$, $CH_3O_3S^-$, $Mg^{2+}$, and $Cl^-$ ranged within 10-80 ng m$^{-3}$ (Table 3).

The predominant elements were Ca ($0.42 \pm 0.33$ µg m$^{-3}$), K ($0.13 \pm 0.08$ µg m$^{-3}$), and Fe ($88 \pm 65$ ng m$^{-3}$), followed by Zn (34
$\pm 33$ ng m$^{-3}$), Pb ($18 \pm 19$ ng m$^{-3}$), Sn ($52 \pm 37$ ng m$^{-3}$), Ti ($5 \pm 4$ ng m$^{-3}$), Ba ($9 \pm 13$ ng m$^{-3}$), Sr ($2 \pm 5$ ng m$^{-3}$). Mn, Ni, Cr, and
Se concentrations were below $2 \pm 1$ ng m$^{-3}$. Tracer metals such as Ti, Cr, Mn, K, Ca, Fe, Ni, Cu, Zn Sr, Pb and Se were found
in all $PM_{2.5}$ samples, while V was not found in any sample. Other tracer metals such as As and Sb were detected only at a
reduced number of samples with concentrations below 20 ng m$^{-3}$. Table 3 shows the mean, standard deviation, minimum and
maximum concentration of the carbonaceous fraction, soluble ions, and metals found in the $PM_{2.5}$ samples collected in CRV.














Table 3. Mean, 1 standard deviation, minimum and maximum concentrations of carbonaceous fraction, soluble ions, and
metals in samples of $PM_{2.5}$ collected in Palmira.

| Species | | Mean | SD | Min | Max | Units |
|---|---|---|---|---|---|---|
| $PM_{2.5}$ | | 14.38 | 4.35 | 6.73 | 24.45 | $\mu g\ m^{-3}$ |
| OC | | 3.97 | 1.31 | 2.31 | 8.35 | |
| EC | | 0.96 | 0.31 | 0.52 | 2.15 | |
| $SO_4^{-2}$ | | 2.15 | 1.39 | 0.98 | 10.27 | |
| $NH_4^+$ | | 0.67 | 0.62 | 0.18 | 4.29 | |
| $NO_3^-$ | | 0.51 | 0.30 | 0.11 | 1.45 | |
| $Na^+$ | | 0.21 | 0.16 | 0.02 | 0.45 | |
| $Ca^{+2}$ | (Water soluble ion) | 0.14 | 0.06 | 0.06 | 0.28 | |
| $C_2O_4^{-2}$ | | 0.11 | 0.06 | 0.04 | 0.36 | |
| $K^+$ | (Water soluble ion) | 0.09 | 0.06 | 0.02 | 0.30 | |
| Ca | (Trace metal) | 0.42 | 0.33 | 0.01 | 1.95 | |
| K | (Trace metal) | 0.13 | 0.08 | 0.02 | 0.46 | |
| Formate | | 82 | 88 | 0 | 217 | $ng\ m^{-3}$ |
| $PO_4^{-3}$ | | 66 | 42 | 10 | 148 | |
| Methansulfonate | | 50 | 36 | 13 | 256 | |
| Cl- | | 20 | 19 | 0 | 75 | |
| $Mg^{+2}$ | | 19 | 10 | 2 | 52 | |
| $NO_2^-$ | | 3 | 1 | 1 | 6 | |
| Fe | | 88 | 64 | 2 | 293 | |
| Sn | | 52 | 37 | 9 | 137 | |
| Zn | | 34 | 33 | 0 | 153 | |
| Pb | | 18 | 19 | 0 | 84 | |
| Ba | | 9 | 13 | 2 | 72 | |
| Sb | | 8 | 5 | 3 | 22 | |
| Cu | | 6 | 5 | 1 | 22 | |
| Ti | | 5 | 4 | 0 | 17 | |
| As | | 2 | 4 | 0 | 10 | |
| Mn | | 2 | 1 | 0 | 5 | |
| Ni | | 2 | 1 | 0 | 9 | |
| Sr | | 2 | 5 | 0 | 28 | |
| Cr | | 1 | 1 | 0 | 4 | |
| Se | | 1 | 1 | 0 | 6 | |
| V | | 0 | 1 | 0 | 3 | |






**3.3.  PM₂.₅ mass closure**

The mass closure (Figure 3) shows the crucial contribution of organic material (52.99% ± 17.79%) and the secondary inorganic
fraction, represented by ammoniated sulphate (16.12 ± 3.98%) and ammonium nitrate (3.19 ±1.71%). EC constituted 6.95 ±
2.52% of PM₂.₅. The mineral fraction corresponded to dust (8.67 ± 5.71%) and TEO (0.82 ± 0.44%). The sea salt was 0.80 ±
1.28 % and PBW 5.20 ± 1.20%. A mass closure of 93.40 ± 33.38% was achieved. Although the PM₂.₅ concentrations observed
in the CRV were not so high as compared with those registered in Brazil and Mexico during the preharvest season, the EC
percentage is in a similar range or slightly lower than those observed in other urban areas (Snider et al., 2016), showing the
key role of incomplete combustion processes in the area.

The average (OC/EC) ratio found in CRV was 4.2 ± 0.72, from which we can infer that secondary aerosol formation had a
relevant role. The segregation of OC in the primary and secondary fraction was made using the EC tracer method applied in
previous studies (Pio et al., 2011; Plaza et al., 2011).  The $(OC/EC)_{min}$ ratio selected to differentiate $OC_{prim}$ from $OC_{sec}$ was the
minimum ratio observed, equivalent to 2.12. Still, this value could induce the overestimation of $OC_{prim}$ due to the distance
between the emission sources and the sampling site (27 m overground), and by the local meteorological conditions that favor
the volatilization and oxidation of organic components into particles before being collected. As result, $OC_{prim}$ was estimated
as 50.3% and $OC_{sec}$ as 49.7 % over the total OC, with a minimum variability of 3.8%. The estimated $OM_{pri}$ concentration was
2.95 ± 1.05 µg m⁻³ and the $OM_{sec}$ concentration was 4.08 ± 1.86 µg m⁻³ that represented the 24.4 and 31.0 of PM₂.₅ respectively.

The mineral fraction, quantified as the sum of the oxides present in the crustal material (dust) and other trace element oxides
(TEO) contributed 9.1 ± 5.5% and 0.9 ± 0.4%, respectively. Despite the non-quantification of highly abundant mineral dust
elements such as Si, the concentrations of Ca, Ti, and Fe indicated the impact of soil resuspension on the PM₂.₅ mass
concentration.

Particle-bound water (PBW) depends on the concentration of hygroscopic compounds embodied in the particulate matter and
relative humidity of the weighing room where the PM₂.₅ mass collected on the filters was determined. In this study, it was
assumed that (i) $NH_4^+$, $SO_4^{2-}$ and $NO_3^-$ were the main compounds responsible for the absorbed water and (ii) thermodynamic
equilibrium is dominated by these ions that allow calculating the $H^+$ molar fraction as a difference of $(SO_4^{2-} + NO_3^-)$ and $NH_4^+$
required to establish the charge neutrality. Polar organic compounds and other water-soluble ions were not considered in the
present study. The PBW content was estimated using the mean measured concentrations of $NH_4^+$, $SO_4^{2-}$ and $NO_3^-$ in the AIM
Model, where a multiplier factor was found equivalent to 0.32 as a proportion between the concentrations of summatory of
theses ions and the water fraction contained in the PM₂.₅. As a result, the PBW was 5.3% of PM₂.₅ mass concentration.



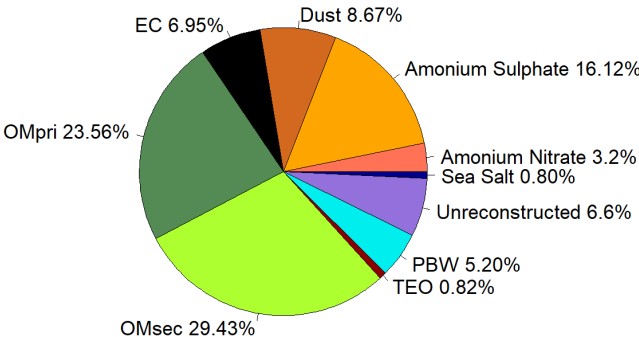


386            Figure 3. Mean fractions of PM$_{2.5}$ components of in the CRV.

### 3.4.    Ions

Anion- and cation-equivalent (AE and CE, respectively) charges were compared to estimate the acidity of PM$_{2.5}$ (Figure 4). AE and CE displayed a tight Spearman linear correlation ($r^2$=0.99). The AE to CE ratio of $1.2 \pm 0.1$ suggests that cations were generally well balanced by anions and that PM$_{2.5}$ was nearly neutral. Just a few samples displayed AE/CE ratios significantly higher than 1, i.e. slightly acidic, which might be attributed to the sulfate dianion ($SO_4^{2-}$) abundance. The ratio between the two main water-soluble ions, ammonium cation ($NH_4^+$) and $SO_4^{2-}$, was $[NH_4^+]/[SO_4^{2-}] = 0.3 \pm 0.1$ This indicates that fine PM in CRV is more acidic than suggested by the AE/CE ratio. This acidity might be explained by insufficient ammonium in CRV's atmosphere to neutralize $SO_4^{2-}$ present in fine particulate matter.

Sulfate to nitrate ratios ($[SO_4^{2-}]/[NO_3^-]$) have been used as indicators of the relative contribution of mobile and stationary sources to particulate matter nitrogen and sulfur (Agarwal et al., 2020; Begam et al., 2016). High ratio values indicate dominance of stationary sources over vehicular emissions. The measured average ratio of $[SO_4^{2-}]/[NO_3^-] = 4.5 \pm 2.9$ indicates that stationary sources are predominant in CRV. This ratio is higher than the one reckoned from measurements in Brazil by Souza et al. (2014) at Piracicaba ($3.6 \pm 1.0$) and Sao Paulo ($1.8 \pm 1.0$). The strong correlations between $SO_4^{2-}$ and $NH_4^+$ ($r^2 = 0.84$), $SO_4^{2-}$ and methanesulfonic acid ($CH_3O_3S^-$) ($r^2 = 0.47$), and $SO_4^{2-}$ and oxalate dianion ($C_2O_4^{2-}$) ($r^2 = 0.57$) allows to infer that inorganic secondary aerosol formation is a significant PM$_{2.5}$ source in CRV. In addition, the presence of potassium cation ($K^+$) in submicron particles is recognized as a biomass burning tracer (Andreae, 1983; Ryu et al., 2004). $K^+$ showed a moderate correlation with nitrite anion ($NO_2^-$) ($r^2 = 0.56$) and $C_2O_4^{2-}$ ($r^2$=0.54) in CRV, which suggests that biomass burning influences secondary aerosol formation. $Mg^{2+}$ and $Ca^{2+}$ ions, usually considered crustal metals, exhibited a moderate correlation of $r^2 = 0.59$ (Li et al., 2013). Also, $Mg^{2+}$ and $C_2O_4^{2-}$ moderate correlation ($r^2 = 0.42$) points to a link among crustal species and secondary aerosols. Such an association could be plausibly explained by soil erosion induced by pyro-convection during sugarcane pre-harvest burning (Wagner et al., 2018). Our study full species correlation matrix is shown in Fig 3S.


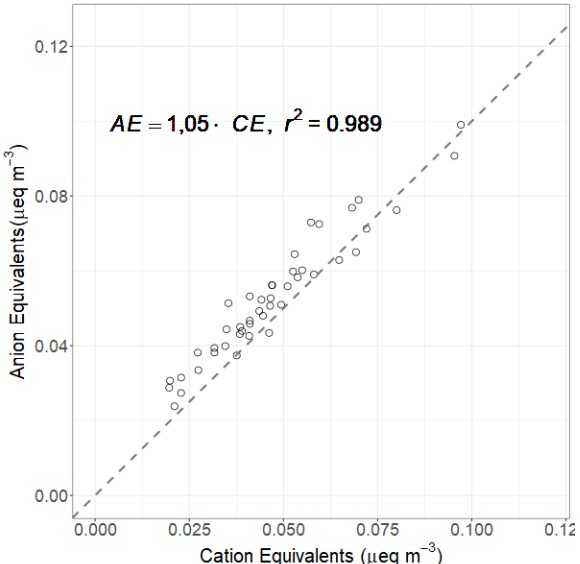


Figure 4. Scatter plot of the equivalent cations and anions for PM$_{2.5}$ samples collected in Palmira.
**3.5.  Metals**

The measured total PM$_{2.5}$ trace metal concentration was 706 ± 462 ng m$^{-3}$ (101.3 ng m$^{-3}$ to 2638 ng m$^{-3}$). Trace metals can
originate from non-exhaust and exhaust emissions. The non-exhaust emissions come from brake and tire wear, road surface
abrasion, wear/corrosion of other vehicle components, and resuspension of road surface dust. Exhaust emissions metals are
related to fuel, lubricant combustion, catalytic converters, and engine corrosion. As shown by Kundu and Stone (2014), many
of these sources share some metals in their chemical composition profile, , thus an unambiguous specific source attribution is
non-trivial. In this study, we found a significant correlation among Fe, Mn and Ti (r$^2$ ≈ 0.72), which is typically associated
with high abundance of crustal material (Fomba et al., 2018), and substantiates the importance of soil dust as a significant
source in CRV. Also, tire and brake wear tracer metals, including Zn and Cu, showed weaker but still significant correlations
among them (r$^2$ ≈ 0.32). PM$_{2.5}$ Ca concentrations at Palmira were quite high (405 ± 334 ng m$^{-3}$ (1.6 ng m$^{-3}$ to 1952 ng m$^{-3}$).
These levels can be attributed to dust generation by agricultural practices, particularly land planning, liming and tilling, PHB
pyro-convection induced soil erosion, and traffic-induced soil resuspension on unpaved rural roads. One of the very few
previous investigations on PM composition in CRV (Criollo and Daza, 2011) analyzed trace metals in PM$_{10}$ at 4 CRV locations,
including Palmira. They found significant enrichment of Fe and K metals at locations exposed to PHB. It must be bear in mind
that PM$_{10}$ samples included coarse mode aerosols, of which dust might have been a significant fraction. Also, environmental
regulations have been successful in steadily reducing the sugarcane burned area in CRV since 2009. Burned area dropped from
72% in 2011 to 35.46% in 2018, our year of measurements (Cardozo-Valencia et al., 2019).



Cd, Pb, Ni, Hg and As, and other metals and metalloids are considered carcinogenic (WHO Regional Office for Europe, 2020).
Measured concentrations of Pb and Ni in $PM_{2.5}$ at Palmira were 18 ng m$^{-3}$ (+/-19) and 2 ng m$^{-3}$ (+/-1), respectively. These
mean values were below the EU target value (0.5 µg m$^{-3}$ and 20 ng m$^{-3}$ respectively) (WHO, 2013a), and below the allowed
annual limit of the Colombian national air quality standard (0.5 µg m$^{-3}$ and 0.18 µg m$^{-3}$ respectively) (MADS, 2017). These
concentrations are nevertheless significantly higher than those reported for other suburban areas in Midwestern United States
and remote sites in northern tropical Atlantic (Fomba et al., 2018; Kundu and Stone, 2014). Pb concentrations are similar to
those reported for Bogota and other large urban areas (SDA, 2010; Vasconcellos et al., 2007). Pb has been long banned as fuel
additive in Colombia thus the observed levels might be associated with metallurgical industry and waste incineration.
Information on ambient air hazardous metal concentration in Latin America urban and rural areas is still scarce.

**3.6. Carbohydrates**

Levoglucosan is a highly specific biomass burning organic tracer. Along with K$^+$, OC and EC, it can be used to effectively
identify the relevance of biomass burning as aerosol source. The relative contribution of levoglucosan to the particulate matter
carbohydrate burden, and specially the levoglucosan to mannosan ratio, can be used as indicators of type of biomass burned
(Engling et al., 2009). In this study, the following carbohydrates were quantified: levoglucosan, mannosan, glucose, galactosan,
fructose and arabitol. Levoglucosan was by far the most abundant (113.8 ± 147.2 ng m$^{-3}$), reaching values of up to 904.3 ng
m$^{-3}$, followed by glucose (10.4 ± 6.1 ng m$^{-3}$), mannosan (7 ± 6.1 ng m$^{-3}$), and arabitol (4.1 ± 3.5 ng m$^{-3}$). Levoglucosan and
mannosan were detected in all $PM_{2.5}$ samples, while galactosan and fructose were detected only in a very reduced number of
samples. Levoglucosan accounted for 3.5±2.3% of OC and 0.96% ± 0.81% of $PM_{2.5}$.

The levoglucosan concentration found in this study was quite similar to the reported in areas of Brazil where sugarcane
production and processing are important economic activities. For instance, during the harvest (*zafra*) period in Araraquara, the
levoglucosan mean concentration was 138 ± 91 ng m$^{-3}$, although during the non-harvest period was unexpectly high (73 ± 37
ng m$^{-3}$) (Urban et al., 2014). Likewise, the levoglucosan average concentration at Piracicaba during a reduced fire period was
66 ng m$^{-3}$ (Souza et al., 2014). The measured mean levoglucosan/mannosan ratio in Palmira was 17.6 ± 13.0 (min: 8.1 – max:
58.1). Chemical profile studies found a levoglucosan/manossan ratio of ~10 for sugarcane leaves burned in stoves (Hall et al.,
2012; Dos Santos et al., 2002) and of ~54 for burned bagasse (Dos Santos et al., 2002). Leaves constitute the largest fraction
(20.8%, Victoria et al., 2002) of pre-harvest burned sugarcane. Consistently and expectedly, the levoglucosan/mannosan ratio
at Palmira is much closer to the chemical profile ratio of leaves than that of bagasse. Moreover, ambient air samples in
Araraquara and Piracicaba showed levoglucosan/mannosan ratios of 9 ± 5 and ~33, respectively. For comparison, the
levoglucosan/mannosan ratio in particulate matter from rice straw and other crops burning were ~26.6 and~23.8, respectively
(Engling et al., 2009). This indicates that the levoglucosan/manossan ratio is sensitive to the type of biomass burned but also



to burning conditions. The large levoglucosan/mannosan ratio variability in our study suggest that Palmira was impacted by
sugarcane pre-harvest burning most of the time but also by bagasse combustion in sugar mills to a lesser extent. Levoglucosan
and mannosan emissions factors from bagasse combustion have not been reported so far. We hypothesize that, even if these
were very small, levoglucosan and mannosan combustion emissions might not be negligeable as CRV sugarcane biomass
yields are very high and most of the harvested sugarcane bagasse is combusted for electric power and steam production.

**3.7.      Polycyclic Aromatic Hydrocarbons (PAH)**

A total of 22 PAHs were measured in each sample collected at Palmira, including the 16 PAHs listed as human health priority
pollutants by WHO and US-EPA (Yan et al., 2004). The total PAH concentration was $5.6 \pm 2.9$ ng m$^{-3}$ (min: 2.3 ng m$^{-3}$ – max:
15.8 ng m$^{-3}$). Figure 5a shows the PAH concentration variability during the sampling campaign (mean and standard deviation
are available on Table S2). The most abundant PAH were FLE (44.2%±11.9% total concentration share), ANT (9,10)
(10.0%±4.5%), BbF (7.4%±2.3%), BghiP (6.7%±2.4%), IcdP (6.4%±1.9%), CPY (6.0%±2.3%), FLO (9H) (5.4%±3.1%),
BeP(4.6%±1.3%), and BaP(4.4%±1.6%), which accounted for 95.1% of the total PAH concentration(Figure 5b). Three-ring
PAHs were the most abundant (59.04% of total PAH). Put together, five- and six- ring PAHs accounted for an additional
38.44%. The less abundant PAH group was the four-ring (2.52%). A previous study in CRV, carried out by Romero et al.
(2013), but on PM$_{10}$ samples, showed higher FLT, PYR and PHE concentrations in areas highly exposed to sugarcane pre-
harvest burning compared to other locations. In contrast, PM$_{2.5}$ FLE concentrations in this research were significantly higher
than those in PM$_{10}$ by Romero et al. (2013), while PYR and PHE levels were similar .

The carcinogenic species BaP, BbF, BkF, BaA, BghiP, FLE, CPY and BeP were identified in all the PM$_{2.5}$ samples. BaP is a
reference for PAH carcinogenicity (WHO, 2013a) that is used as PAH exposure metrics, known as the BenzoaPyrene-
equivalent carcinogenic potency (BaPE). We calculated BaPE using the toxic equivalent factors (TEF) proposed by Nisbet
and LaGoy (1992) and (Malcolm and Dobson, 1994). PAH concentrations were multiplied by TEF and then added to estimate
the carcinogenic potential of PM$_{2.5}$-bounded PAH. The mean carcinogenicity level at Palmira, expressed as BaP-TEQ, was 0.4
± 0.2 ng m$^{-3}$ (min: 0.1 ng m$^{-3}$ - max: 1.4 ng m$^{-3}$). Only one sample exceeded the Colombian annual limit of 1 ng m$^{-3}$ but most
of them exceeded the WHO reference level of 0.12 ng m$^{-3}$. The mutagenic potential of PAH (BaP-MEQ) was estimated using
the mutagenic equivalent factors (MEF) reported for Durant et al., (1996). The average BaP-MEQ was 0.5 ± 0.3 ng m$^{-3}$ (min:
0.2 ng m$^{-3}$ - max: 1.8 ng m$^{-3}$). These levels are comparable to those measured in PM$_{2.5}$ by Mugica-Álvarez et al., (2016) in
Veracruz (México) but during the sugarcane non-harvest period. PM$_{10}$ BaP-MEQ levels in Araraquara (Brazil) (de Andrade et
al., 2010; De Assuncao et al., 2014) were twice as high as those found in. This suggest that year-long sugarcane pre-harvest
burning in CRV leads to lower mutagenic potentials compared to those at locations where the harvesting period is shorter
(*zafra*) thus with higher burning rates. We estimated the average BaP-TEQ and BaP-MEQ concentrations in CRV according



to their exposure to sugarcane burning products from Romero et al., (2013) data and used as a benchmark to our measurements,.
$PM_{10}$-bound BaP-TEQ and BaP-MEQ levels for areas not directly exposed to sugarcane burning were 0.16 ng m$^{-3}$ and =0.21
ng m$^{-3}$, respectively. Toxicity and mutagenicity due to $PM_{10}$-bound PAHs were a factor 4 higher at areas directly exposed to
sugarcane burning. It is reasonable to assume that PAHs are largely bound to fine aerosol (<2.5 µm), thus that our
measurements are comparable to (Romero et al., 2013). If so, our site at Palmira would be at an intermediate exposure
condition, higher than areas not directly exposed to sugarcane burning but lower than exposed zones.

Ratios among different PAHs have been extensively used to distinguish between traffic and other PAH sources. We used the
diagnostic ratios presented by Ravindra et al. (2008) and Tobiszewski and Namieśnik (2012a) to better understand the
contribution of sources to $PM_{2.5}$ in CRV. The benzo(e)pyrene ratio to the sum of benzo(e)pyrene and benzo(a) pyrene is used
as an aerosol aging indicator. Local or "fresh" aerosols have [BeP]/([BeP]+[BaP]) ratios around 0.5, while aged aerosols can
have ratios as low as cero as a result of photochemical decomposition and oxidation. The [BeP]/([BeP]+[BaP]) ratio at Palmira
was $0.51 \pm 0.04$, with a majority (84.4%, n = 38) of fresh samples a minor fraction (15.6%, n=7) of photochemically-degraded
samples.

Other two diagnostic ratios were used to assess the prevalence of traffic as $PM_{2.5}$ source. The first ratio one used IcdP and
BghiP, two automobile emissions markers (Miguel and Pereira, 1989). Values higher than 0.5 for the IcdP ratio to the sum of
IcdP and BghiP, [IdcP]/([IdcP]+[BghiP]), indicate aged particles (Tobiszewski and Namieśnik, 2012) generated by coal, grass
or wood burning (Yunker et al., 2002). The second ratio is [BaP]/[BghiP]. Ratios higher than 0.6 are indicative of traffic
emissions (Tobiszewski and Namieśnik, 2012). . At Palmira, the [IdcP]/([IdcP]+[BghiP]) and [BaP]/[BghiP] ratios were 0.48
$\pm$ 0.04 and $0.69 \pm 0.13$, which indicates that ~63% of the samples originated from combustion of oil products (n = 30), and
~36% came from non-traffic sources, like wood, grass, or coal (n = 15).

Also, the structure and size of PAHs are indicative of their sources. PAHs with low molecular weight (LMW) (two or three
aromatic rings) has been reported as tracers of wood, grass and fuel oil combustion, while the PAHs of medium molecular
weight (MMW) (four rings) and high molecular height (HMW) (five and six rings) are associated with coal combustion and
vehicular emissions. The ratio between LMW ratio to the sum of MMW and HMW, LMW/(MMW+HMW), is used for source
identification. Ratios lower than one are indicative oil products combustion, while ratios larger than one are associated to coal
and biomass combustion (Tobiszewski and Namieśnik, 2012). The ratio at Palmira, LMW/(MMW+HMW) = $1.43 \pm 1.00$, was
rather variable but suggests that a large fraction of PAHs in CRV (82.2% of samples) were generated by biomass burning or
combustion, as coal combustion is quite limited nowadays. Just one in five samples (17.8%) have PAHs attributable to oil
products combustion.





Sugarcane-burning emitted PAHs are mainly of low molecular weight, especially of two (~66% of PAHs) and three rings
(~27%), among which FLE, PHE and ANT are the most emitted, according to Hall et al. (2012) chemical profile. The relative
abundance of three-ring PAHs (Figure 5) in CRV's PM$_{2.5}$ is likely due open-field sugarcane pre-harvest burning to major
extent and to controlled bagasse combustion for electric power and steam production to a lesser extent.

The highest concentrations of PAH were observed on 10$^{th}$ August and 11$^{th}$ September 2018 with levels of 15.8 ng m$^{-3}$ and 14.4
ng m$^{-3}$, respectively (Fig 5S). In particular, on 10$^{th}$ August 2018 elevated concentrations of 5 and 6 rings PAHs were observed.
a change in wind circulation pattern form previous day on (Fig S2), with a wind speed reduction and a predominance of winds
from the north. Then on 11$^{th}$ September 2018, we observed an increase of 3-ring PAHs and winds from NW at the average
wind speed at the sampling location.

This indicates that there were at least two types of sources. The abundance of HMW PAHs indicate fossil fuel combustion
sources, and LMW PAHs suggest that parts of these came from non-fossil fuel combustion sources.

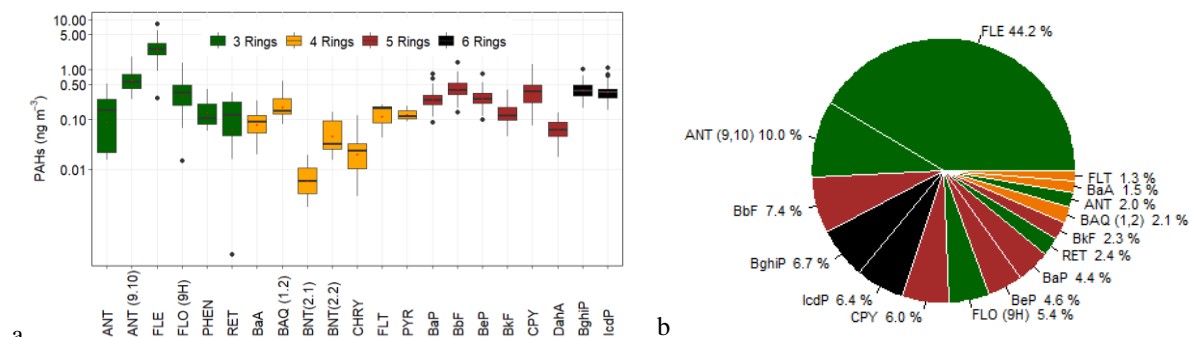

Figure 5. Abundance of PAHs measured in PM$_{2.5}$ samples collected in CRV, represented by colors according to the number
of rings of each PAH, green (tree rings), yellow (four rings), brown (five rings) and black (six rings). a) Box-plot of
concentrations in ng m$^{-3}$, red dots represent mean concentrations of each PAH. b) pie-plot of the relative abundance of PAHs
in PM$_{2.5}$ samples.
**3.8.    Alkanes**

A total of 16 alkanes ranging from C$_{20}$ up to C$_{34}$ were analyzed in this study and used to identify the presence of fossil fuel
combustion and plant fragments in the PM$_{2.5}$ samples. The abundance of total n-alkanes during the whole sampling period was
in the range of 13.0 to 88.45 ng m$^{-3}$ with an average concentration of 40.36 ng m$^{-3}$ ± 18.82 ng m$^{-3}$. In general, the high molecular
weight n-alkanes such as C$_{29}$ − C$_{31}$ were the most abundant. These are characteristic of vegetative detritus corresponding to
plant fragments in airborne particle matter (Lin et al., 2010). The most abundant n-alkanes were C$_{29}$, C$_{30}$ and C$_{31}$ (Fig 6.).





Likewise, the carbon number maximum concentration ($C_{max}$) was $C_{29}$ in 43% of samples and $C_{31}$ in 28% of them. This result
is consistent with the chemical profile of sugarcane burning reported by (Oros et al., 2006) with $C_{max}$ of $C_{31}$.

The carbon preference index (CPI) and wax n-alkanes percentage (WNA%) are parameters used to elucidate the origin of the
n-alkanes and infer whether emissions come from biogenic or anthropogenic sources. The CPI represents the ratio between
odd and even carbon number n-alkanes. The equation used to calculate CPI in the present study is shown in Table 2, following
the procedure reported by (Marzi et al., 1993). Values of CPI ≤ 1 (or close to 1) indicate that n-alkanes are emitted from
anthropogenic sources, while values higher than 1 indicate the influence of vegetative detritus in the $PM_{2.5}$ samples (Mancilla
et al., 2016). In this study, mean CPI was always greater than 1, with an average value of $1.22 \pm 0.18$ (min:1.02 – max:1.8)
that is between the CPI for fossil fuel emissions of ~1.0 (Caumo et al., 2020) and sugarcane burning of 2.1 (Oros et al., 2006),
revealing the influence of several sources over the $PM_{2.5}$ in the CRV.

Likewise, WNA% represents the preference of odd n-alkanes in the sample. The odd n-alkanes, especially of higher molecular
weight, are representative of plant wax related emissions. The waxes are present on the surface of plants, especially on the
leaves, and they become airborne by a direct or indirect mechanism like wind action or biomass burning (Kang et al., 2018;
Simoneit, 2002). In this research, the samples analyzed showed a preference for odd carbon on $C_{27}$, $C_{29}$, $C_{31}$ and $C_{33}$, which
have higher concentrations than the next higher and lower even carbon number homologs, proving the biogenic contribution
over the $PM_{2.5}$ in the CRV. The WNA% was calculated using the equation shown in Table 2 described by Yadav et al. (2013).
A larger WNA% represents the contribution from emissions of plant waxes or biomass burning. Otherwise, a smaller value
represents that n-alkanes from petrogenic sources, known as petrogenic n-alkanes (PNA)%. The mean WNA% calculated for
the $PM_{2.5}$ samples collected from the CRV was $12.65\% \pm 5.21\%$ (min: 4.71% – max: 29.92%) and can be defined as petrogenic
inputs (PNA%) that were 87.35% during the sampling period. The correlation between CPI and WNA was moderate ($r^2$=0.53)
supporting a consistent meaning between these two parameters, and they are useful for assessing the plant wax contribution
on $PM_{2.5}$.

Overall, the total concentration of n-alkanes of the $PM_{2.5}$ in the CRV was lower than those reported in areas where the sugarcane
is often burned in Brazil (Urban et al., 2016), although the behavior of the parameters of CPI and $C_{max}$ is similar. Compared
with other urban areas in Latin American, the n-alkane concentration in the CRV was similar to that reported in the
metropolitan zone of the Mexican valley (MZMV) for $PM_{2.5}$ (Amador-Muñoz et al., 2011), and Bogota for $PM_{10}$ and slightly
lower than reported in Sao Paulo for $PM_{10}$ (Vasconcellos et al., 2011). However, the CPI and WNA in these cities were smaller
than in the CRV, because of the strong influence of vehicular emissions in these densely populated cities. The OC/EC ratio
was moderately associated with WNA values ($r^2 = 0.41$), indicating that an increase of this ratio can be explained by the
vegetative detritus contribution in the $PM_{2.5}$, while the levoglucosan concentrations did not show correspondence to the CPI





and WNA values; therefore, the levoglucosan levels did not explain the preference of odd carbon number homologs. These
results indicated that n-alkanes found in this study came from several sources with a noticeable contribution from plant wax
emissions. The parameters used to assess the source contribution of PM$_{2.5}$ through n-alkanes such CPI and WNA%, were
characteristic of aerosols collected in urban areas.

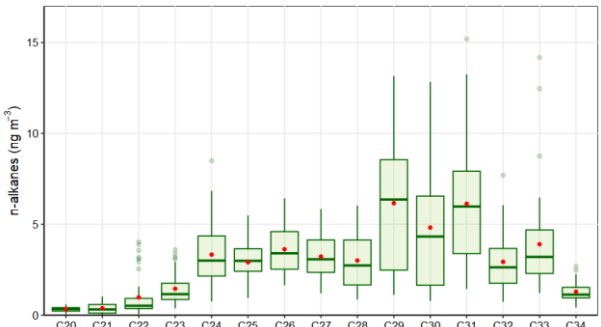


592                    Figure 6. Average n-alkanes concentrations in PM$_{2.5}$ samples

**3.9.    PCA**

We applied a PCA to the chemical composition data to assess the latent factors controlling the PM$_{2.5}$ concentrations in the
CRV. This statistical tool was used to find the chemical species that describe each component and qualitatively associate these
to potential sources of fine aerosol particles. In order to extract the number of components in a PCA many procedures exist,
while one of the most common ones is the scree plot of successive eigenvalues for several components from which it is possible
to identify the point where the proportion of the variance explained by each subsequent component drops off abruptly. Fig S7
shows the inflection point in component number four, explaining 45% of the chemical composition data variance. The addition
of two following components allows describing 61% of the variance. Therefore, this study was conducted taking into account
six components. Table 4 shows the loading for each chemical component assessment for the six components, where the
loadings higher than 0.6 were considered in the discussion interpreted as a source that contributed to the formation of PM$_{2.5}$.









Table 4. Loading of PCA after varimax rotation. Loading with |x| < 0.2 was considered insignificant and removed, while
loading |x| < 0.6 is considered high and is **printed bold.**

| Principal Component | PC1 Road dust resuspension | PC2 Secondary aerosols and biomass burning | PC3 Fuel Combustion 1 | PC4 Detritus vegetables | PC5 Fuel Combustion 2 | PC6 Agricultural soil resuspension |
|---|---|---|---|---|---|---|
| % variance explained | 13 | 11 | 11 | 10 | 9 | 7 |
| % cumulative variance | 13 | 24 | 35 | 45 | 54 | 61 |
| OC | 0.26 | **0.66** | 0.34 | 0.33 | 0.34 | 0.2 |
| EC | | **0.73** | 0.39 | | 0.37 | |
| $C_2O_4^{2-}$ | | **0.65** | | 0.52 | | 0.39 |
| $K^+$ (Water soluble ion) | | 0.39 | | **0.69** | | 0.3 |
| $NO_2^-$ | | **0.65** | 0.23 | 0.43 | | |
| $F^-$ | | 0.32 | | 0.36 | | |
| $NO_3^+$ | | 0.41 | | | | **0.64** |
| $Cl^-$ | | | | | | **0.61** |
| $Ca^{2+}$ (Water soluble ion) | | 0.29 | | | | **0.74** |
| $NH_4^+$ | | **0.87** | | | | |
| $SO_4^{2-}$ | | **0.88** | | | | 0.26 |
| Formate | | | | 0.34 | | 0.25 |
| $Na^+$ | | 0.33 | | | 0.25 | 0.36 |
| $Mg^{2+}$ | | 0.24 | 0.21 | | | **0.77** |
| Methansulfonate | | **0.81** | | 0.23 | | |
| $PO_4^{3-}$ | | 0.38 | | | | |
| Cr | | | | | | |
| Ca | **0.63** | | | | 0.2 | 0.41 |
| K | 0.43 | 0.2 | | **0.62** | 0.2 | |
| Ti | 0.39 | | | | | |
| Fe | **0.74** | 0.35 | | | | |
| Sb | 0.58 | | | | | |
| Mn | 0.46 | 0.26 | | | | |
| Ba | **0.73** | | 0.25 | | | |
| Se | 0.40 | 0.55 | | | | |
| Zn | **0.80** | | | | | |
| As | | 0.57 | | | | |



| | | | | | | | |
|---|---|---|---|---|---|---|---|
| Sr | 0.37 | | | | | | |
| Pb | 0.24 | 0.42 | | | | | |
| Sn | **0.61** | | 0.25 | | | 0.35 | |
| Cu | 0.52 | 0.24 | 0.3 | | | | |
| Ni | 0.3 | 0.25 | | | | | |
| C20 | | | 0.33 | | | 0.29 | 0.29 |
| C21 | 0.51 | | 0.29 | | | 0.32 | |
| C22 | | | | | **0.85** | | |
| C23 | 0.3 | | | | **0.84** | | |
| C24 | | | 0.43 | 0.25 | **0.67** | | |
| C25 | | | | | **0.84** | | |
| C26 | | | 0.34 | 0.33 | **0.78** | | |
| C27 | | | 0.29 | 0.53 | **0.65** | | |
| C28 | | | 0.24 | | **0.62** | 0.48 | 0.26 |
| C29 | | | | | **0.69** | 0.33 | 0.26 |
| C30 | | | | | **0.68** | 0.32 | 0.21 |
| C31 | | | | | **0.78** | 0.3 | |
| C32 | | | | | **0.76** | | |
| C33 | | | | | **0.79** | | |
| C34 | | | | | **0.65** | | |
| FLT | | | | | | | |
| PYR | | | | | | | |
| BNT (2,1) | | | | | | | |
| RET | 0.33 | | | | | | |
| PHEN | | 0.46 | | | | 0.28 | |
| BAQ (1,2) | | | | | | | |
| ANT | | 0.47 | | | | 0.27 | |
| DahA | 0.48 | | 0.42 | | | | |
| CPY | 0.32 | | **0.76** | 0.21 | | | |
| BaA | | 0.21 | **0.8** | | | | |
| IcdP | | 0.23 | **0.75** | | | | |
| BghiP | 0.28 | | **0.74** | 0.37 | | | |
| BkF | 0.32 | | **0.71** | 0.37 | | 0.23 | |
| BbF | 0.22 | 0.28 | **0.83** | 0.27 | | | |
| BeP | 0.27 | 0.21 | **0.8** | 0.32 | | | |
| BaP | | | **0.83** | 0.34 | | | |
| CHRY | 0.42 | | 0.47 | 0.32 | | | |
| FLE | | 0.26 | | | | **0.73** | |





| | | | | | | |
|---|---|---|---|---|---|---|
| ANT (9,10) | | | | 0.2 | 0.25 | 0.2 |
| FLO (9H) | | | 0.24 | 0.44 | | 0.42 |
| BNT (2.2) | | | 0.48 | 0.2 | | |
| Glucose | | 0.51 | | | 0.22 | |
| Galactosan | | 0.31 | | 0.34 | 0.2 | 0.23 |
| Levoglucosan | 0.27 | 0.41 | | 0.31 | | |
| Arabitol | | **0.6** | | 0.3 | | |
| Fructose | 0.38 | | 0.29 | | | |


The first component rotated (PC1) explained 13% of the total variance in the dataset. PC1 exhibited high loadings for the
metals Zn, Fe, Ba, Ca, Sn, and a minor loading for Sb, Cu, Mn, K. These metals could have their origins in road dust
resuspension because of roadside particles contained in non – exhaust and exhaust car emissions. For instance (Pant and
Harrison, 2013) have shown the emission of Zn and Ca from tire wear and  Fe, Ba, Cu, Sb, and Sn from brake wear. Also, this
component explained the variance of the n-alkane $C_{21}$ and a variance proportion of the HMW PAH (DahA, CPY, BkF, BghiP,
BbF) associated with vehicular emissions, together with Cu (Miguel and Pereira, 1989). Therefore, we call PC1 a component
associated to road dust resuspension.

The second rotated component (PC2) explained 11% of the total variance. PC2 is a component associated with secondary
aerosol formation and biomass burning. It was described by high loadings of the ions $SO_4^{2-}$, $NH_4^+$, methansulfonate, $C_2O_4^{2-}$,
and $NO_2^-$, along with a fraction of EC and OC. Those ions also are observed in another region with sugarcane preharvest
burning in Brazil (Allen et al., 2004), where the plume is enriched with $Cl^-$, $NO_3^-$ and $Na^+$ in the fine fraction of aerosol
particles, while the ions $SO_4^{2-}$ and $C_2O_4^{2-}$ are formed in the atmosphere during transport process due to the oxidation of $SO_2$
and hydrocarbons. The important fraction of the variance of OC and especially EC explained by PC2 indicated the effect of
an incomplete combustion process on this component, which together with the variance proportion explained by the
Levoglucosan and $K^+$ indicated that the combustion process was associated with biomass burning. Also, PC2 is the one that
best explained the variance of PAHs FLE and ANT,  abundant in the chemical profile of sugarcane burning particles (Hall et
al., 2012; Simoneit, 2002). Thus, PC2 seemed to be a combined effect of secondary aerosol formation and sugarcane burning.

The third rotated component (PC3) explained 11% of the variance and has high loadings for the PAH: BbF, BaP, BeP, BaA,
CPY, IcdP, BghiP, BkF, and the n-alkane $C_{20}H_{42}$ typically emitted during incomplete combustion of vehicle fuels (Andrade et
al., 2012; Miguel and Pereira, 1989). A similar fraction of the variance of OC and EC was also explained with PC3, supporting
the contribution of the combustion process. Thus, PC3 could be interpreted as a component derived from petroleum emissions
by traffic.


The fourth rotated component (PC4) explained 10% of the variance and had high loadings for n-alkanes >$C_{27}$, $K^+$ and K. The
n-alkanes >$C_{25}$ are frequently associated with detritus and vegetable waxes. We explained the emission of the higher molecular
weight n-alkanes to the biomass present in the region used by the agriculture industry and the abundance of nature present in
the CRV. Therefore, we named PC4 a component associated with detritus vegetables.

The fifth rotated component (PC5) explained 9% of the variance, where $C_{22}$ to $C_{27}$ alkanes had high loadings. These fractions
were associated with anthropogenic emissions (Kang et al., 2018). Thus, PC5 could be interpreted as a component derived
from anthropogenic emissions. In addition, PC5 explained a variance proportion of some species associated with vehicular
engine combustion, such as BghiP, BkF, BbF, characteristics of gasoline vehicles (Kuo et al., 2013) joint to EC and OC
derivates from incomplete combustion. In summary, the components PC2, PC3 and PC5 describe the variance of EC, meaning
the impact of incomplete combustion present in the region.

The sixth rotated component (PC6) explains 7% of the variance exhibiting high loadings for $Mg^{2+}$, $Ca^{2+}$, FLE, $NO_3^+$, and $Cl^-$
and moderate for others such as $C_2O_4^{2-}$, $Na^+$, $K^+$. Particularly, the ions $Cl^-$, $NO_3^+$, $K^+$ increase during biomass burning (Ryu et
al., 2004). In addition, PC6 strongly explained the variance of calcium as water soluble ions and a fraction of the trace metal,
therefore the erosion of soil could be considered as an activity that explained PC6. After preharvest biomass burning and fires
as a tool to prepare the land for the next crops the soil erosion can increase because of the reduction of vegetation. Therefore,
compounds associated with soil erosion and derivates of biomass burning can simultaneously affect the soil erosion and the
chemical composition of $PM_{2.5}$.

The PCA results showed there was no dominant component that explained the variance of chemical species contained in $PM_{2.5}$
in CRV. Instead, many components have roles equally important that are associated with road dust derived from traffic, the
formation of secondary aerosol particles and biomass combustion, petroleum combustion associated with vehicular exhaust,
and the presence of vegetative detritus and agriculture soil resuspended by wind erosion. Sugarcane burning was not identified
as an individual component that can be explained because the open sugarcane burns happened continuously during the
sampling, so they became a background source for this study that very likely was included in the secondary formation as
another background source. However, the carbohydrates contained in $PM_{2.5}$ was linked to the characteristic species of
secondary aerosol formation and vegetative detritus. Therefore the secondary pollutants could also originate from the burning
of sugarcane in the CRV, similar to the results reported by (Vasconcellos et al., 2007) in Brazil.




## 4.  Conclusions

PM$_{2.5}$ samples collected in the Cauca River Valley (CRV), Colombia, were analyzed to determine the main chemical components of fine aerosol particles and to qualitatively identify aerosol sources using ratios and principal component analysis (PCA). The main PM$_{2.5}$ components were organic material (52.99%here), followed by ammonium sulfate (16.12%here) and elemental carbon (6.95%here). The contribution of secondary organic material and inorganic salts was found to be significant and likely related to biomass burning and agricultural practices and estimated secondary aerosol formation was estimation of. EC and PAHs concentrations confirm the presence of incomplete combustion process in CRV. Diagnostic ratios applied to organic compounds indicate that PM$_{2.5}$ was emitted locally and had contributions of pyrogenic and petrogenic sources. In addition, levoglucosan and mannosan levels showed that biomass burning was ubiquitous during the sampling period. Fluoranthene (FLE) was the most abundant PAH, confrieming the strong influence of sugarcane burning. Five- and six-ring PAH associated with vehicular emissions were also abundant in PM$_{2.5}$. Our measurements point to sugarcane pre-harvest burning as the main source of PAHs in CRV. The comparison of PM$_{2.5}$ concentrations and mutagenic potentials suggest that year-long sugarcane pre-harvest burning in CRV, which is also conducted on less than half of the harvested area (34% in 2018) and over limited plots sizes (~6 ha median), leads to lower atmospheric pollutant burdens and mutagenic potentials compared to those at locations where the harvesting period is shorter (*zafra*) thus with higher burning rates.

Several sources were identified through PCA, including road dust, secondary aerosol particles, and biomass combustion, vehicle exhaust, vegetative detritus and resuspended agricultural soil likely induced by pre-harvest burning. Not one of these sources was dominant nor explained the chemical species variance of measured PM$_{2.5}$. Sugarcane burning was not identified as an independent source, but it was found related to the secondary aerosol formation component on PCA. This link between sugarcane burning emissions and secondary aerosol formation requires further investigation. We found that the effects of agriculture on CRV's air quality, particularly of sugarcane preharvest burning are non-trivial. Besides primary particles, this activity generates SOA precursors, induces soil resuspension and is closely tied to diesel emissions during harvesting.

*Author contribution:* RJ, GR-S, and NR conceived and managed the project. LM-F, ACV-B, GR-S, and RJ set the instruments up and performed the aerosol sampling. LM-F carried out the sample chemical analysis at TROPOS with the guidance and support of DvP, MvP, KW and HH. LM-F and ACV-B analyzed the measurement results, including PCA and other techniques with the support of DvP and RJ. LM-F, RJ, NR and ACV-B prepared the manuscript with substantial contributions from all the authors.

*Competing interests:* The authors declare that they have no conflict of interest.



*Acknowledgments:*  The authors gratefully acknowledge the financial support from the Universidad Nacional de Colombia –
Sede Palmira ([Impacto de la quema de caña de azúcar en la calidad del aire del Valle Geografico del Río Cauca] CACIQUE
project Hermes # 37718) and Leibniz Institute for Tropospheric Research (TROPOS) for analytical support. This project was
supported by EU granted the mobility project PAPILA. We thank Susanne Fuchs, Anke Roedger, Sylvia Haferkorn, and
Kornelia Pielok for their technical assistance in the chemical analysis of samples.  We acknowledge Pablo Gutierrez for his
contributions in the processing of open sugarcane burning base data.

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

Contributions to cities ' ambient particulate matter ( PM ): A systematic review of local source contributions at global level,
Atmos. Environ., 120, 475–483, https://doi.org/10.1016/j.atmosenv.2015.08.087, 2015.
Khedidji, S., Müller, K., Rabhi, L., Spindler, G., Fomba, K. W., Van Pinxteren, D., Yassaa, N. and Herrmann, H.: Chemical
characterization of marine aerosols in a south mediterranean coastal area located in Bou Ismaïl, Algeria, Aerosol Air Qual.
Res., 20(11), 2448–2473, https://doi.org/10.4209/aaqr.2019.09.0458, 2020.
Kundu, S. and Stone, E. A.: Composition and sources of fine particulate matter across urban and rural sites in the Midwestern
United States, Environ. Sci. Process. Impacts, 16(6), 1360–1370, https://doi.org/10.1039/c3em00719g, 2014.
Kuo, C. Y., Chien, P. S., Kuo, W. C., Wei, C. T. and Rau, J. Y.: Comparison of polycyclic aromatic hydrocarbon emissions
on gasoline- and diesel-dominated routes, Environ. Monit. Assess., 185(7), 5749–5761, https://doi.org/10.1007/s10661-012-

812    2981-6, 2013.

Lara, L. L., Artaxo, P., Martinelli, L. A., Camargo, P. B., Victoria, R. L. and Ferraz, E. S. B.: Properties of aerosols from
sugar-cane    burning    emissions    in    Southeastern    Brazil,    Atmos.    Environ.,    39(26),    4627–4637,
https://doi.org/10.1016/j.atmosenv.2005.04.026, 2005.
Lee, S., Wang, Y. and Russell, A. G.: Assessment of secondary organic carbon in the southeastern United States: A review, J.
Air Waste Manag. Assoc., 60(11), 1282–1292, https://doi.org/10.3155/1047-3289.60.11.1282, 2010.
Li, X., Wang, L., Ji, D., Wen, T., Pan, Y., Sun, Y. and Wang, Y.: Characterization of the size-segregated water-soluble
inorganic ions in the Jing-Jin-Ji urban agglomeration: Spatial/temporal variability, size distribution and sources, Atmos.
Environ., 77, 250–259, https://doi.org/10.1016/j.atmosenv.2013.03.042, 2013.
Lin, L., Lee, M. L. and Eatough, D. J.: Review of recent advances in detection of organic markers in fine particulate matter
and their use for source apportionment, J. Air Waste Manag. Assoc., 60(1), 3–25, https://doi.org/10.3155/1047-3289.60.1.3,

823    2010.

Lopez, M. E. and Howell, W. E.: Katabatic winds in the Equatorial Andes, J. Atmos. Sci., 24(1), 29–35,
https://doi.org/doi.org/10.1175/1520-0469(1967)024<0029:KWITEA>2.0.CO;2, 1967.
Lyu, R., Shi, Z., Alam, M. S., Wu, X., Liu, D., Vu, T. V., Stark, C., Xu, R., Fu, P., Feng, Y. and Harrison, R. M.: Alkanes and
aliphatic carbonyl compounds in wintertime PM2.5 in Beijing, China, Atmos. Environ., 202(November 2018), 244–255,
https://doi.org/10.1016/j.atmosenv.2019.01.023, 2019.
MADS: Res. No 2254, Ministerio de Ambiente y Desarrollo Sostenible, Colombia., 2017.
Majra, J. P.: Air Quality in Rural Areas, in Chemistry, Emission Control, Radioactive Pollution and Indoor Air Quality,





https://doi.org/10.5772/16890, , 2011.
Malcolm, H. M. and Dobson, S.: The calculation of an Environmental Assessment Level (EAL) for atmospheric PAHs using
relative potencies., 1994.
Mancilla, Y., Mendoza, A., Fraser, M. P. and Herckes, P.: Organic composition and source apportionment of fine aerosol at
Monterrey, Mexico, based on organic markers, Atmos. Chem. Phys., 16(2), 953–970, https://doi.org/10.5194/acp-16-953-

836  2016, 2016.

Marzi, R., Torkelson, B. E. and Olson, R. K.: A revised carbon preference index, Org. Geochem., 20(8), 1303–1306,
https://doi.org/10.1016/0146-6380(93)90016-5, 1993.
Mesa S., Ó. J. and Rojo H., J. D.: On the general circulation of the atmosphere around Colombia, Rev. la Acad. Colomb.
Ciencias Exactas, Fis. y Nat., 44(172), 857–875, https://doi.org/10.18257/RACCEFYN.899, 2020.
Miguel, A. H. and Pereira, P. A. P.: Benzo(k)fluoranthene, benzo(ghi)perylene, and indeno(1, 2, 3-cd)pyrene: New tracers of
automotive  emissions  in  receptor  modeling,  Aerosol  Sci.  Technol.,  10(2),  292–295,
https://doi.org/10.1080/02786828908959265, 1989.
Mugica-Alvarez, V., Santiago-de la Rosa, N., Figueroa-Lara, J., Flores-Rodríguez, J., Torres-Rodríguez, M. and Magaña-
Reyes, M.: Emissions of PAHs derived from sugarcane burning and processing in Chiapas and Morelos México, Sci. Total
Environ., 527–528, 474–482, https://doi.org/10.1016/j.scitotenv.2015.04.089, 2015.
Mugica-Álvarez, V., Ramos-Guízar, S., Santiago-de la Rosa, N., Torres-Rodríguez, M. and Noreña-Franco, L.: Black Carbon
and Particulate Organic Toxics Emitted by Sugarcane Burning in Veracruz, México, Int. J. Environ. Sci. Dev., 7(4), 290–294,
https://doi.org/10.7763/ijesd.2016.v7.786, 2016.
Mugica-Álvarez, V., Hernández-Rosas, F., Magaña-Reyes, M., Herrera-Murillo, J., Santiago-De La Rosa, N., Gutiérrez-
Arzaluz, M., de Jesús Figueroa-Lara, J. and González-Cardoso, G.: Sugarcane burning emissions: Characterization and
emission factors, Atmos. Environ., 193, 262–272, https://doi.org/10.1016/j.atmosenv.2018.09.013, 2018.
Neusüss, C., Pelzing, M., Plewka, A. and Herrmann, H.: A new analytical approach for size-resolved speciation of organic
compounds in atmospheric aerosol particles: Methods and first results, J. Geophys. Res. Atmos., 105(D4), 4513–4527,
https://doi.org/10.1029/1999JD901038, 2000.
Nisbet, I. C. T. and LaGoy, P. K.: Toxic equivalency factors (TEFs) for polycyclic aromatic hydrocarbons (PAHs), Regul.
Toxicol. Pharmacol., 16(3), 290–300, https://doi.org/10.1016/0273-2300(92)90009-X, 1992.
Oros, D. R., Abas, M. R. bin, Omar, N. Y. M. J., Rahman, N. A. and Simoneit, B. R. T.: Identification and emission factors of
molecular tracers in organic aerosols from biomass burning: Part 3. Grasses, Appl. Geochemistry, 21(6), 919–940,
https://doi.org/10.1016/j.apgeochem.2006.01.008, 2006.
Ortiz, E. Y., Jimenez, R., Fochesatto, G. J. and Morales-Rincon, L. A.: Caracterización de la turbulencia atmosférica en una
gran zona verde de una megaciudad andina tropical, Rev. la Acad. Colomb. Ciencias Exactas, Físicas y Nat., 43(166), 133,
https://doi.org/10.18257/raccefyn.697, 2019.



Pan, X., Ichoku, C., Chin, M., Bian, H., Darmenov, A., Colarco, P., Ellison, L., Kucsera, T., Da Silva, A., Wang, J., Oda, T.
and Cui, G.: Six global biomass burning emission datasets: Intercomparison and application in one global aerosol model,
Atmos. Chem. Phys., 20(2), 969–994, https://doi.org/10.5194/acp-20-969-2020, 2020.
Pant, P. and Harrison, R. M.: Estimation of the contribution of road traffic emissions to particulate matter concentrations from
field measurements: A review, Atmos. Environ., 77, 78–97, https://doi.org/10.1016/j.atmosenv.2013.04.028, 2013.
Pereira, G. M., Oraggio, B., Teinilä, K., Custódio, D., Huang, X., Hillamo, R., Alves, C. A., Balasubramanian, R., Rojas, N.
Y. and Sanchez-Ccoyllo, O.: A comparative chemical study of PM 10 in three Latin American cities : Lima, Medellín, ans São
Paulo, Air Qual. Atmos. Heal., 12, 1141–1152, https://doi.org/10.1007/s11869-019-00735-3, 2019.
Pio, C., Cerqueira, M., Harrison, R. M., Nunes, T., Mirante, F., Alves, C., Oliveira, C., Sanchez de la Campa, A., Artíñano, B.
and Matos, M.: OC/EC ratio observations in Europe: Re-thinking the approach for apportionment between primary and
secondary organic carbon, Atmos. Environ., 45(34), 6121–6132, https://doi.org/10.1016/j.atmosenv.2011.08.045, 2011.
Plaza, J., Artíñano, B., Salvador, P., Gómez-Moreno, F. J., Pujadas, M. and Pio, C. A.: Short-term secondary organic carbon
estimations with a modified OC/EC primary ratio method at a suburban site in Madrid (Spain), Atmos. Environ., 45(15), 2496–
2506, https://doi.org/10.1016/j.atmosenv.2011.02.037, 2011.
Ramírez, O., Sánchez de la Campa, A. M., Amato, F., Catacolí, R. A., Rojas, N. Y. and de la Rosa, J.: Chemical composition
and source apportionment of PM10 at an urban background site in a high–altitude Latin American megacity (Bogota,
Colombia), Environ. Pollut., 233, 142–155, https://doi.org/10.1016/j.envpol.2017.10.045, 2018.
Ravindra, K., Sokhi, R. and Van Grieken, R.: Atmospheric polycyclic aromatic hydrocarbons: Source attribution, emission
factors and regulation, Atmos. Environ., 42(13), 2895–2921, https://doi.org/10.1016/j.atmosenv.2007.12.010, 2008.
Rojo H., J. D. and Mesa O., Ó. J.: A simple conceptual model for the heat induced circulation over Northern South America
and Meso-America, Atmosphere (Basel)., 11(11), 1–14, https://doi.org/10.3390/atmos11111235, 2020.
Romero, D., Sarmiento, H. and Pachón, J. E.: Estimación de hidrocarburos aromáticos policíclicos y metales pesados asociados
con la quema de caña de azúcar en el valle geográfico del río Cauca , Colombia, Rev. Épsilon, 21(2013), 57–82, 2013.
Ryu, S. Y., Kim, J. E., Zhuanshi, H., Kim, Y. J. and Kang, G. U.: Chemical composition of post-harvest biomass burning
aerosols in gwangju, Korea, J. Air Waste Manag. Assoc., 54(9), 1124–1137,
https://doi.org/10.1080/10473289.2004.10471018, 2004.
Dos Santos, C. Y. M., Azevedo, D. de A. and De Aquino Neto, F. R.: Selected organic compounds from biomass burning
found in the atmospheric particulate matter over sugarcane plantation areas, Atmos. Environ., 36(18), 3009–3019,
https://doi.org/10.1016/S1352-2310(02)00249-2, 2002.
Schauer, J. J.: Sources contributions to atmospheric organic compound concentrations: Emissions measurments and model
predictions, California Institute Technology, 1998.
SDA: Plan decenal de descontaminación del aire de Bogotá, Bogotá D.C.
http://ambientebogota.gov.co/en/c/document_library/get_file?uuid=b5f3e23f-9c5f-40ef-912a-



51a5822da320&groupId=55886, 2010.
Simoneit, B. R. T.: Biomass burning - A review of organic tracers for smoke from incomplete combustion, Appl.
Geochemistry, 17(3), 129–162, https://doi.org/10.1016/S0883-2927(01)00061-0, 2002.
Snider, G., Weagle, C. L., Murdymootoo, K. K., Ring, A., Ritchie, Y., Stone, E., Walsh, A., Akoshile, C., Anh, N. X.,
Balasubramanian, R., Brook, J., Qonitan, F. D., Dong, J., Griffith, D., He, K., Holben, B. N., Kahn, R., Lagrosas, N., Lestari,
P., Ma, Z., Misra, A., Norford, L. K., Quel, E. J., Salam, A., Schichtel, B., Segev, L., Tripathi, S., Wang, C., Yu, C., Zhang,
Q., Zhang, Y., Brauer, M., Cohen, A., Gibson, M. D., Liu, Y., Martins, J. V., Rudich, Y. and Martin, R. V.: Variation in global
chemical composition of PM2.5: emerging results from SPARTAN, Atmos. Chem. Phys., 16(15), 9629–9653,
https://doi.org/10.5194/acp-16-9629-2016, 2016.
Souza, D. Z., Vasconcellos, P. C., Lee, H., Aurela, M., Saarnio, K., Teinilä, K. and Hillamo, R.: Composition of PM2.5 and
PM10 collected at Urban Sites in Brazil, Aerosol Air Qual. Res., 14(1), 168–176, https://doi.org/10.4209/aaqr.2013.03.0071,

908    2014.

Sutton, M. A., Billen, G., Bleeker, A., Erisman, J. W., Grennfelt, P., Grinsven, H. Van, Grizzetti, B., Howard, C. M. and Leip,
A.: Technical summary Part I Nitrogen in Europe : the present position, in The European Nitrogen Assessment: Sources,
Effects and Policy Perspectives, edited by M. A. Sutton, C. M. Howard, J. W. Erisman, G. Billen, A. Bleeker, P. Grennfelt, H.
Van Grinsven, and B. Grizzetti, Cambridge University Press, Cambridge, https://doi.org/10.1017/CBO9780511976988.003, ,

913    2011.

Szabó, J., Szabó Nagy, A. and Erdős, J.: Ambient concentrations of PM10, PM10-bound polycyclic aromatic hydrocarbons
and heavy metals in an urban site of Győr, Hungary, Air Qual. Atmos. Heal., 8(2), 229–241, https://doi.org/10.1007/s11869-

916    015-0318-7, 2015.

Tobiszewski, M. and Namieśnik, J.: PAH diagnostic ratios for the identification of pollution emission sources, Environ. Pollut.,
162, 110–119, https://doi.org/10.1016/j.envpol.2011.10.025, 2012.
Turpin, B. J. and Lim, H.: Species Contributions to PM2 . 5 Mass Concentrations : Revisiting Common Assumptions for
Estimating    Organic    Mass,    Aerosol    Sci.    Technol.,    35:1(September    2014),    37–41,
https://doi.org/http://dx.doi.org/10.1080/02786820119445, 2001.
Urban, R. C., Lima-Souza, M., Caetano-Silva, L., Queiroz, M. E. C., Nogueira, R. F. P., Allen, A. G., Cardoso, A. A., Held,
G. and Campos, M. L. A. M.: Use of levoglucosan, potassium, and water-soluble organic carbon to characterize the origins of
biomass-burning aerosols, Atmos. Environ., 61, 562–569, https://doi.org/10.1016/j.atmosenv.2012.07.082, 2012.
Urban, R. C., Alves, C. A., Allen, A. G., Cardoso, A. A., Queiroz, M. E. C. and Campos, M. L. A. M.: Sugar markers in aerosol
particles    from    an    agro-industrial    region    in    Brazil,    Atmos.    Environ.,    90(2014),    106–112,
https://doi.org/10.1016/j.atmosenv.2014.03.034, 2014.
Urban, R. C., Alves, C. A., Allen, A. G., Cardoso, A. A. and Campos, M. L. A. M.: Organic aerosols in a Brazilian agro-
industrial    area:    Speciation    and    impact    of    biomass    burning,    Atmos.    Res.,    169,    271–279,





https://doi.org/10.1016/j.atmosres.2015.10.008, 2016.
Vargas, F. A., Rojas, N. Y., Pachon, J. E. and Russell, A. G.: PM10 characterization and source apportionment at two
residential areas in Bogota, Atmos. Pollut. Res., 3(1), 72–80, https://doi.org/10.5094/APR.2012.006, 2012.
Vasconcellos, P. C., Balasubramanian, R., Bruns, R. E., Sanchez-Ccoyllo, O., Andrade, M. F. and Flues, M.: Water-soluble
ions and trace metals in airborne particles over urban areas of the state of São Paulo, Brazil: Influences of local sources and
long range transport, Water. Air. Soil Pollut., 186(1–4), 63–73, https://doi.org/10.1007/s11270-007-9465-2, 2007.
Vasconcellos, P. C., Souza, D. Z., Ávila, S. G., Araújo, M. P., Naoto, E., Nascimento, K. H., Cavalcante, F. S., Dos, M.,
Smichowski, P. and Behrentz, E.: Comparative study of the atmospheric chemical composition of three South American cities,
Atmos. Environ., 45(32), 5770–5777, https://doi.org/10.1016/j.atmosenv.2011.07.018, 2011.
Victoria, J., Amaya, A., Rangel, H., Viveros, C., Cassalett, C., Carbonell, J., Quintero, R., Cruz, R., Isaacs, C., Larrahondo, J.,
Moreno, C., Palma, A., Posada, C., Villegas, F. and Gómez, L.: Características agronómicas y de productividad de la variedad
Cenicaña Colombiana (CC) 85-92, Cali., 2002.
Villalobos, A. M., Barraza, F., Jorquera, H. and Schauer, J. J.: Chemical speciation and source apportionment of fine particulate
matter in Santiago, Chile, 2013, Sci. Total Environ., 512–513, 133–142, https://doi.org/10.1016/j.scitotenv.2015.01.006, 2015.
Wagner, R., Jähn, M. and Schepanski, K.: Wildfires as a source of airborne mineral dust - Revisiting a conceptual model using
large-eddy simulation (LES), Atmos. Chem. Phys., 18(16), 11863–11884, https://doi.org/10.5194/acp-18-11863-2018, 2018.
WHO Regional Office for Europe: Air quality guidelines for Europe, pp. 457–465, World Health Organization, Copenhagen,
Denmark, https://doi.org/10.1525/9780520948068-070, , 2020.
World Health Organization: Review of evidence on health aspects of air pollution - REVIHAAP Project.
http://www.euro.who.int/pubrequest%0Ahttp://www.euro.who.int/__data/assets/pdf_file/0004/193108/REVIHAAP-Final-
technical-report-final-version.pdf., 2013.
Wu, C. and Zhen Yu, J.: Evaluation of linear regression techniques for atmospheric applications: The importance of appropriate
weighting, Atmos. Meas. Tech., 11(2), 1233–1250, https://doi.org/10.5194/amt-11-1233-2018, 2018.
Yadav, I. C. and Devi, N. L.: Biomass burning, regional air quality, and climate change, 2nd ed., Elsevier Inc., 2019.
Yadav, S., Tandon, A. and Attri, A. K.: Monthly and seasonal variations in aerosol associated n-alkane profiles in relation to
meteorological     parameters     in     New     Delhi,     India,     Aerosol     Air     Qual.     Res.,     13(1),     287–300,
https://doi.org/10.4209/aaqr.2012.01.0004, 2013.
Yan, J., Wang, L., Fu, P. P. and Yu, H.: Photomutagenicity of 16 polycyclic aromatic hydrocarbons from the US EPA priority
pollutant     list,     Mutat.     Res.     -     Genet.     Toxicol.     Environ.     Mutagen.,     557(1),     99–108,
https://doi.org/10.1016/j.mrgentox.2003.10.004, 2004.
Yunker, M. B., Macdonald, R. W., Vingarzan, R., Mitchell, H., Goyette, D. and Sylvestre, S.: PAHs in the Fraser River basin:
a critical appraisal of PAH ratios as indicators of PAH source and composition, Org. Geochem., 33, 489–515,
https://doi.org/doi.org/10.1016/S0146-6380(02)00002-5, 2002.