# Peer review of "Understanding aerosol composition in a tropical inter-Andean valley impacted by agro-industrial and urban emissions"

_Atmospheric Chemistry and Physics, 2021_

## Author Response (AR1)

**Response to reviewers**

Manuscript: Understanding aerosol composition in a tropical inter-Andean valley impacted by agro-industrial and urban emissions

Authors: Lady Mateus-Fontecha, Angela Vargas-Burbano, Rodrigo Jimenez, Nestor Y. Rojas, German Rueda-Saa, Dominik van Pinxteren, Manuela van Pinxteren, Khanneh Wadinga Fomba, and Hartmut Herrmann

We thank the referees for their thoughtful and thorough reviews.

The aim of this manuscript is to show and analyze the chemical characterization of $PM_{2.5}$ samples collected in an area where air quality is affected by industrial and agro-industrial activities, as well as anthropogenic emissions that are typical in dense cities. $PM_{2.5}$ in this region is transported through the Cauca River Valley (CRV) and is susceptible to suffer a photochemical transformation process due to high temperature and high relative humidity.

Despite the importance of the region as an agricultural and agro-industrial hub, emission inventories are incomplete and are out of date. Therefore, the contribution of different sources to total emissions and air pollution is still highly uncertain. With this study, we aim to contribute to the understanding of aerosol pollution in the CRV and the influence of different sources, through its chemical speciation and the analysis of diagnostic ratios among chemical species. Source apportionment, however, is out of the scope of this manuscript.

Following reviewers' comments, we have changed the title of the manuscript to: "Understanding aerosol composition in a tropical inter-Andean valley impacted by agro-industrial and urban emissions". The aim of this modification is to remove the focus of the manuscript on sugarcane burning, since it is not the only activity that contributes to particulate air pollution. The revised focus is on analyzing aerosol composition and diagnostic ratios to infer the influence of several sources, including those associated with the sugarcane agro-industry.

**Response to Anonymous Referee #1**

**General comment**

This study evaluated the chemical composition of PM2.5 particles in the Cauca River Valley (CRV), an agro-industrial region in Southwest Colombia where sugarcane production is a common activity. Organic material followed by ammonium sulfate and elemental carbon were found as the main components of the PM2.5 in the region. I consider that the provided information is important for the region and for other countries where sugarcane production is a recurrent practice; however, the current manuscript needs to be significantly improved before it can be accepted for publication. I am wondering if the Editor considers that this manuscript fits into the "Measurement Report" category.

**Authors' response**

We thank reviewer #1 for his/her general comment. We understand that the Editor may change the manuscript type to "Measurement report".

**Major comments**

Although English is not my mother tongue, it is clear that the present manuscript requires to be deeply edited and cleaned as there are several grammatical errors and a significant amount of typos. I encourage the senior coauthors to read the revised manuscript before they submit it.

The authors need to highlight the novelty of the present study. Additionally, a real scientific discussion is required. Currently, the discussion focussed on indicating if the obtained results are lower, higher or comparable to previous results. This is not enough and I invite the authors to do a better job. At the moment this is more like a description of the results with a lack of a scientific discussion.

Redundancy needs to be avoided. The authors define the same term/variable several times along the text. In other cases, after a term/variable is defined the authors do not use the defined abbreviations.

**Authors' response to major comments**

We thank reviewer #1 for his major comments. We have edited the manuscript according to his/her suggestions, improving grammar, correcting typos, defining the terms just in the abstract, when applicable, and in their first appearance, and reducing redundancy. We have improved the discussion of the results.

Minor comments

Line 46: "biomass burning" should be "biomass burning (BB)"
Change was made
Line 63: "emits aerosols of high toxicity". Such as? Please add a reference here. Change was made
Line 63: "secondary organic aerosol precursors". Such as? Please add a reference here. Change was made
Line 63: "short-lived 63 climate pollutants". Such as? Please add a reference here. Change was made
Line 93: "of particulate matter (PM)". Define it the first time it is used. Change was made
Line 95: "organic components". Such as? Change was made
Line 96: "numerous enough". Provide a number. Change was made
Lines 100-101: "putting together disparate source data". What do the authors mean? A complete emission inventory is not available for the whole region. Therefore, a preliminary estimation was made from sources of information with disparate quality. Nevertheless, the sentence was deleted.
Line 110: "This pollutant source". PM2.5? Change was made
Lines 113-114: "including elemental carbon (EC), primary and secondary organic carbon (OC)". Already defined in lines 74 and 75. Change was made
Line 115: PAH was used previously in line 75. Change was made
Line 118: "pre-harvest burning". It was used several times before. Change was made
Line 131-132: "2.2 Mhab". It was mentioned in line 83. Change was made
Line 132: "123 khab". In line 83 129 khab was mentioned. Change was made
Line 133: "across the Western Cordillera , as shown in Figure 1. on the Pacific Ocean coast is one most". Something is wrong here. Change was made
Line 152: "Particulate matter" should be "Particulate matter (PM)" Change was made
Line 153: "filters were stored". Please add more details such as the containers. Change was made
Line 154: "1888 sugarcane pre-harvest burning events". How was this quantified? The source was the Regional Environmental Authority, which receives it from sugar companies. Change was made in the manuscript
Line 154 and along the text: "pre-harvest burning" should be "PHB" Change was made
Lines 169-170: "were analyzed from Teflon and quartz filters". In line 159 it is mentioned that the metals were obtained from the Teflon filters only. Changes were made to both sentences
Lines 171-172: "Metals were analyzed from three 8-mm circular pieces punched from the 45 filters". As far as i know, Teflon filters cannot be cut in small pieces. How did you do this?

The method followed to prepare the samples used to identify and quantify metals was described by Wadinga Fomba et al., (2020). We used a ceramic puncher to extract two spots of 8-mm in diameter from Teflon filters and another ceramic puncher for the quartz filters. We added the reference to the manuscript.

Line 178: "and polycyclic aromatic hydrocarbons (PAH)". Already defined. Change was made

Line 178: "circular pieces of filter". Please clarify the type of filter you are referring to. Change was made

Line 189: "blank filters". This was not mentioned before. Can the authors elaborate more on how those filters were collected/treated/processed? We added a description of the procedure followed for blank filters.

Line 194: "elemental carbon (EC)". Already defined. Change was made

Line 198: "equations (Chow et al., 2015). See Table 1. Also, this reconstruction". Something is wrong here. Change was made

Line 200: "particle bounded water" should be "PBW" This is the first time the term appears, so it was left as "particle-bounded water (PBW)"

Lines 282-283, 285: "(Lopez and Howell, 1967)". Is this the up to date knowledge? We added a more recent reference.

Line 286: Add a reference after "basis"? Is there something recent to show that this applies nowadays? Reference was added

Line 290: "first at 14.5 m". Above the ground? Yes. We added the term "above the ground"

Line 299: "were very likely controlled by solar radiation". What other mechanisms could work in the region?

Line 442: "Levoglucosan is a highly specific biomass burning organic tracer". Add a reference and explain why. We added a reference.

Line 449: "samples". How many? We added the sample number for each species.

Lines 463-464: "The large levoglucosan/mannosan ratio variability in our study suggest that Palmira was impacted by sugarcane pre-harvest burning most of the time". I sugges to add a time series of PM2.5 and the levoglucosan levels? We had made a mistake in the text. It is the large magnitude of the levoglucosan/mannosan ratio and not its variability that suggests that Palmira was impacted by sugarcane PHB most of the time. ´We thank the referee's suggestion to add a time series of PM2.5 and levoglucosan. The figure was added to the manuscript.

Figure 1S. How was this information obtained? We added a description of the information source (based on data provided by the CVC, the environmental agency of the CRV, compiled from sugar mills PHB events reports)"

Figure 2. Indicate the time period used to build these figures, is it the sampling period as indicated in the figure caption of it includes the 1 previous year as mentioned in line 290. We added the time periods to the figure caption.

Table 3. "Why there are no decimals? What is the resolution of the used method? We added the decimals that were missing and references that contain detailed information about the methods.

Table 3: I suggest adding a column indicating the number of samples on which each element/ion was detected. We thank the reviewer for this suggestion. We added the number of samples

Table 3 (last arrow). How can it be possible as in line 334 it is mentioned that "while V was not found in any sample" We had made a mistake. V was found in a few samples. We have corrected the sentence.

Figure 4: I do not find this a key figure to be in the main text We thank the reviewer's suggestion. We have moved the figure to supplementary material.

Technical comments

Line 41: Add a reference after "areas". We restructured the Introduction and removed the sentence.

Line 42: Add additional references to Majra (2011). More references were added

Line 45: I doubt that Majra (2011) is the only and the pioneering study reporting this. More references were added

Line 45: Line 41: Add a reference after "(SOA)". References were added

Line 47: Add additional references to Yadav and Devi (2019). More references were added

Line 49: Add additional references to Sutton et al. (2011). More references were added

Line 52: Add a reference after "factors". We restructured the Introduction and removed the sentence.

Line 43: Add a reference after "exposed". We restructured the Introduction and removed the sentence.

Line 55: Add a reference after "Australia". Change was made

Line 61: Add a reference after "USA". Change was made

Line 62: Add a reference after "worldwide". The sentence was omitted from the introduction after it was reorganized.

Line 64: Add a reference after "contaminants". The sentence was omitted from the introduction after it was reorganized.

Line 78: Add a reference after "CRV". Change was made

Line 84: Add a reference after "plantations".

Line 88: Add a reference after "owners". Change was made

Line 89: Add a reference after "biofuel". Change was made

Line 124: Add a reference after "sparce". Change was made

Line 134: Add a reference after "Colombia". References were added

Line 135: Add a reference after "countries". References were added

Line 201: "described by (Clegg et al., 1998)" should be "described by Clegg et al. (1998)" Change was made

Line 225: "As per Table 1". This is not correct. Change was made

Line 229: "out site". Fix it. Change was made

Line 232: Add a reference after "aerosols". The reference is Turpin and Lim, (2001a), which is at the beginning of the sentence.

Line 276: "The Cauca River Valley (CRV)". Already defined. Change was made

Lines 276-277: " valley at ~985 m altitude located ~120 km from the Pacific Ocean". Already mentioned. Change was made

Lines 278 and 281: "(Rojo H. and Mesa O., 2020)" should be "(Rojo and Mesa, 2020)" Change was made

Line 280: Add a reference after "conditions". References were added

Line 287: "this time period". Do the authors mean during the day? That's right, we're talking about during the day.

Line 289: Add a reference after "out". References were added

Line 318: Add a reference after "Brazil". References were added

Line 321: Add a reference after "Mexico". Reference was added

Line 326: "NO3" Fix it. We had made a mistake. A change was made.

Line 333: "Tracer". Trace? A change was made

Line 355: "by ammoniated sulphate". Fix it. We restructured this session. The change was made.

Line 359: Add additional references to Snider et al. (2016). References were added

Line 369: "24.4 and 31.0". Units? Change was made

Lines 371-372: "trace element oxides (TEO)". Already defined. Change was made

Line 376: "Particle-bound water (PBW)". Already defined. Change was made

Line 415: Add a reference after "dust". Amended as suggested

Line 417: "profile, , thus an". Fix it. Amended as suggested

Line 419: "substantiates"??? We have modified the sentence.

Line 425: "It must be bear in mind"??? We have modified the sentence.

Line 465: "have not been reported so far". Where?

Line 674: "(52.99%here)" and "(16.12%here)". Fix it. Change was made

Line 675: "(6.95%here)". Fix it. Change was made

Line 676: "practices and estimated secondary aerosol formation was estimation of." Something is wrong here. We removed these sentences. Change was made

**Response to Anonymous Referee #2**

Major comments

Comment 1

While the data and methods are presented quite well, the manuscript seems unfocused in the introduction. As the abstract makes pretty clear, this is first and foremost an observational report from a rural/suburban area with some AG burning thrown in. So I would expect a discussion of similar studies both in South America as well as other rural areas. Instead, the intro is focused on sugarcane burning, which might be one motivation for this study but is certainly not its main goal nor result. So a reorganization of the intro would I think make clearer what this effort is about and also help shorten the overall length.

Authors Response:

The introduction was reorganized and the focus on sugarcane burning was changed. The introduction now includes information about the diversity of $PM_{2.5}$ sources existing in CRV and the main studies on $PM_{2.5}$ chemical characterization conducted in the region. The literature review shows that studies with a detailed chemical composition on $PM_{2.5}$ in Latin America are scarce. Most of the studies found have been focused on main cities and a few other areas, but there is an important gap in knowledge on $PM_{2.5}$ composition and sources agro-industrial areas.

Comment 2

The paper clearly shows that there is plenty of primary aerosol in the CRV, both from fossil fuel (FF) combustion and agricultural burning (BB). However, the EC/OC ratios for these two types of sources are quite different (also b/c of some fraction of brown carbon getting assigned to EC in the sunset instrument). So using a simple regression to differentiate primary and secondary OC in this case is likely not very accurate, as the authors concede. A multivariate analysis is often superior in separating the different contributions, but the PCA analysis shown later does not really seem to be providing a meaningful distinction. So assuming a better multivariate analysis is not available, it might be better to try to predict primary OC based on a simple linear model that uses e.g. levoglucosan or K+ (which as a non-volatile tracer is likely a better choice) as a predictor for primary BBOM, and the sum of the lower alkanes as a predictor for primary OM from FF. As a boundary condition, we know that for most types of primary BBOA, levoglucosan is 3-6% of the total OA, so 5-8% of OM (see e.g. Jolleys et al 2015 and Sullivan et al 2019). This approach will likely

underestimate secondary OM, but it will possibly give us more insight on what conditions actually lead to these primary components being maximized.

Authors Response:

Attending the suggestion of the reviewer, a simple linear model was applied to find the proportion of $OC_{prim}$ from three sources that are significant in the CRV, namely fossil fuel combustion ($OC_{FF}$), biomass burning ($OC_{BB}$), and vegetable detritus ($OC_{det}$). $OC_{FF}$, $OC_{BB}$ and $OC_{det}$ was quantified using a linear model from the following tracers: BghiP and IcdP for fossil fuel; levoglucosan for biomass burning; and the sum of the highest molecular weight alkanes ($C_{27} - C_{33}$) for vegetable detritus. As a result, we estimated that 16.4% of OC can be attributed to fossil fuel, 15.2% to biomass burning, and 1.5% to vegetable detritus. In total, 32.7% of OC can be attributed to primary OC from these sources. This result is consistent with the apportionment of $OC_{prim}$ obtained by the OC/EC method, which indicated that around 50% of OC can be attributed to primary emissions. This approach does not allow us to infer $OC_{sec}$ because we don't have information about specific organic tracers of this carbonaceous fraction, such as α-pinene, β-caryophyllene, naphthalene-derived, and isoprene-derived SOA. Therefore, we decided to use both methods in the manuscript, to estimate both $OC_{prim}$ and $OC_{sec}$, and check the consistency of OCprim results.

Comment 3

The values currently used for the OM to OC conversion are reasonable, and clearly contribute to a decent mass closure (which I think should be better documented by e.g. showing a timeseries of the mass closure for each filter). However, I am not following the reasoning behind some of the values discussed. For primary BB emissions, in his recent review, Andreae (2019) used 1.6, which is a number that is roughly consistent with the results from other recent primary BB studies (e.g. Hodshire et al, 2019). BBOA can oxidize quite quickly, so secondary BBOA can indeed exhibit much higher OA:OC ratios, but that's not relevant in that context. Likewise, levoglucosan is an excellent tracer for primary BB, and hence does not belong into a discussion of secondary OM/OC. Importantly, references for the 2.1 value used for secondary OM are completely missing.

Authors Response:

We agree with the reviewer that levoglucosan does not belong into a discussion of $OC_{sec}$ and its conversion to $OM_{sec}$. Accordingly, we have removed the corresponding sentence. We have rewritten the paragraph to clarify the OM/OC ratios used and their references. Now it reads: "Turpin and Lim (2001a) recommended an OM/OC ratio of 1.6 ± 0.2 for urban aerosols, and 2.1 ± 0.2 for non-urban aerosols, values comparable with those found

by Aiken et al. (2008), of 1.71 (1.41 – 2.15), where lower values (1.6 – 1.8) are attributed to ground measurements in the morning, and higher values (1.8 – 1.9) to aircraft sample measurements. BB aerosols can have even higher $f$ values (2.2-2.6), due to the presence of organic components with higher molecular weights, e.g., levoglucosan. However, Andreae (2019) recommends a factor of 1.6 for fresh BB aerosol, which is consistent with Hodshire et al (2019). We believe that traffic and biomass burning are the dominant $OC_{prim}$ sources at our site. Therefore, we used $f_1 = 1.6$ to estimate $OM_{pri}$. We used a factor of 2.1 to estimate $OM_{sec}$ from the $OC_{sec}$ fraction. This factor was chosen based on i) recommended ratios of 2.1±0.2 for aged aerosols."

**Comment 4**

I am puzzled by the discussion of particle acidity. It is obviously always problematic to infer acidity without taking into account the gas phase, but having said that, a Cation:Anion ratio of ~0.8 is in most cases indicative of a fairly acidic aerosol (e.g. not that different from the SE US, where pH~1 was reported by Guo et al, 2015). And as the authors write, the NH4:SO4 ratio suggest that the fine aerosol might be even more acidic (Cation:Anion ratio of ~0.6) and that sea salt and dust are probably biasing the bulk ratio high. This has a few important implications that I think should be discussed at greater length.

Authors Response:

We decided to estimate pH for each sample from the concentrations of ions $NH_4^+$, $NO_3^-$, $SO_4^{2-}$, $Cl^-$ and $Na^+$ using the E-AIM thermodynamic model. The estimated pH~2.5 indicates that the aerosol are slightly acid as was previously suggested from the $[NH_4^+]/[SO_4^{2-}]$ ratio, as well as cations/anions equivalent ratios. However, these ratios and pH were not well correlated ($r^2 = $ ), which would indicate that those ratios are not a robust indicator of aerosol acidity, as mentioned by (Pye et al., 2020). The $[NH_4^+]/[SO_4^{2-}]$ ratio observed in the CRV was similar to that reported by Guo et al., (2015) in Southeastern US. However, the pH in the CRV aerosol was significantly higher (pH = 2-4) than that in Southeastern US (pH < 1) (Pye et al., 2020). The pH in the CRV aerosol can be explained by the presence of sulfate and organic compounds (pH = 1-3), as well as some influence of BB aerosol (pH = 4-5). This was added to the manuscript in section 3.3.

**Comment 5**

If the aerosol is this acidic. nitrate is mostly going to partition to the gas phase, which is essentially what the data shows (although as mentioned below, some volatilization might play a role here). So nitrate from both NOx emissions from primary combustion processes

is not going to be detected, and therefore there is no point in interpreting NO3/SO4 ratios as indicative of any particular source. Furthermore, as discussed later in the PCA section, a significant fraction of nitrate seems to be associated with dust, so it is likely that there is no secondary NH4NO3 in this dataset at all. If so, the mass closure might improve by assuming all nitrate to be Ca(NO3)2 and not NH4NO3.

Authors' Response:

We added a discussion about the partitioning of total nitrate ($NO_3^-$ + $HNO_3$) and total ammonium ($NH_4^+$ + $NH_3$) based on the acidity of aerosols in section 3.3. In this case, the aerosol acidity indicates that nitrate volatilization as $HNO_3$ to the gas phase is favored, which is consistent with the lower concentration of $NO_3^-$ (compere with $SO_4^{2-}$). The [$NH_4^+$]/[$SO_4^{2-}$] ratio (~1.6) indicates an environment with insufficient concentrations of ammonium to neutralize $SO_4^{2-}$ completely. Instead, sulfates seem to be partially neutralized as letovicite ([$NH_4$]$_3$H[$SO_4$]$_2$), as suggested by (Lee et al., 2008). Due to the estimated ammonium deficit, we suggest that nitrate salts were present in PM$_{2.5}$ samples as $KNO_3$, $NaNO_3$, or $Ca(NO_3)_2$.

The use of [$SO_4^{2-}$]/[$NO_3^-$] ratio to infer the origin of PM$_{2.5}$ was removed.

Comment  6

But more importantly, given that both agriculture and BB are large sources of NH3(g), this is a very surprising result and I would encourage the authors to try to understand how it comes about. Given the overall composition, a fairly large source of SO2/H2SO4 would be needed to sustain such an imbalance (e.g. like the large SO2 emissions from coal fired plants in the SE US). But that does not seem to be the case here. Alternatively, the marine aerosol coming off the Walker Circulation might be quite acidic (e.g Nault et al, 2021), but that would imply that most of the sulfate is of marine origin, which seems unlikely.

Authors Response:

In order to explain the high concentrations of $SO_4^{2-}$ observed in PM$_{2.5}$, we added a description of the diversity of sources of sulfur-containing compounds ($SO_2$, $SO_3$ and $H_2S$) in the CRV, including coal in power plants and industrial facilities, the biomass burning activities, the emission of $H_2S$ in poultry production animal production system, and the consumption of diesel for vehicles and tractors used to collect the sugarcane harvest (see section 3.3). The high concentration of $SO_4^{2-}$ and its low correlation with typical soluble ions associated with sea salt aerosol, suggests that Walker Circulation is not the main source of sulfate.

**Comment 7**

Related to the previous point, I wonder if there are published MSA/SO4 (or SO4/seasalt) ratios for the coastal areas in Colombia that could be used to address one of the larger open questions in this study, namely how much background aerosol from the ocean (and also the mountains when the katabatic winds kick in) contributes to the air quality in the CRV.

Authors Response:

$PM_{2.5}$ chemical composition studies in the region are scarce and none of this has reported the concentration of methanesulfonate until now. We used the [methanesulfonate]/[SO$_4^{2-}$] ratio to infer the impact of biogenic sulfur compounds on the total SO4-2 levels. $23.6 \pm 20.1$ showing the abundance or between Na+ and was

**Comment 8**

In its current form, the PCA analysis does not really contribute much to our understanding of the data, especially since the factors are not really discussed in the context of the rich marker analysis introduced before. But regardless how it is presented, while the factors for FF combustion make some intuitive sense, that's not the case for the others. E.g. having all the secondary and marine aerosols in one factor (factor 2) probably mostly reflects that all of these come from the same wind direction and have no strong correlations with other factor. Likewise, I was surprised to see levoglucosan in the road dust factor and Cl in one of the dust factors, but not associated with MSA/Na. To be clear, multivariate analysis is often messy, so some oddities are normal, but overall this analysis seems unfinished and tacked on. As the other reviewer already mentioned, this could at least partially be due to PCA not being the best tool to analyze this type of data, especially given how many low S/N tracers are included in the input matrix. But regardless of the choice of method, the whole point of such an analysis is to simplify the trends in the data, and it clearly fails at that. So I would suggest either eliminating it completely, and instead spend more time discussing how to combine the different findings from the marker analysis (which is currently missing), or try to find a new solution, either with PCA or other techniques (e.g. PMF) that is more suitable. In the end, the dataset might just be too flat to get good factors out of it, so the marker discussion approach seems more promising.

Authors Response:

Following the suggestion of Referee #2, we removed the PCA section from the manuscript. In fact, the preliminary PCA solution did not allow us to draw useful conclusions regarding the significance of sources in the CRV. We recognize that PCA is an ordinary least square

fit of parametric variables, that does not include the uncertainty of each measurement and its variability, as indicated by Hopke, (2015). Therefore, its applicability is very limited in comparison with more robust techniques such as PMF or CBM. We will address source contribution analysis in a future study, so we focused on chemical species concentrations and diagnostic ratios in this manuscript.

Minor/technical comments:

All the abstract level numbers seem to be at least 1-2 digits over their actual significance, consider revising. The number of digits was adjusted

All Figures (except maybe Figure 3): please increase the font of the axis by at least 30-50%, very hard to read. The font size in figures was increased

A dust fraction of 9% in PM2.5 is quite high, and would suggest that PM10 is quite a bit larger than PM2.5, this should probably be mentioned in the discussion of inventories vs measurements. The re-calculated result for this fraction, with the adjustments made, was 3.5%, which is less than half of the previous value. Regarding emission inventories,

Semivolatile components of PM, such as ammonium nitrate and SVOA can easily evaporate from filters, biasing the fractional and overall composition measurement (see e.g. Heim et al, 2020 for a recent example). The sampling site seems to have been fair enough removed from primary sources that this is probably a minor concern, but it is a common filter artifact that should be mentioned.

Given how stable the RH was during sampling (which should probably be better emphasized when discussing this), there is nothing wrong with estimating a constant particulate water contribution with E-AIM, although it might be simpler and more comparable to just use speciated kappa values instead (e.g. Peters et al 2007, Brock et al 2016).

I am not too familiar with the CPI measure, but I wonder if BB emission could lead to e.g. a bias towards non-FF emissions, please discuss. The CPI for fossil fuel combustion emissions is around 1, whereas for sugarcane burning is around 2. The occurrence of BB will produce a value higher than 1 for CPI in ambient aerosol, and it would be close to 2 for areas heavily impacted by BB. In this study, a value of 1.2 suggests a moderate influence of BB and vegetative detritus in ambient aerosol.

L44-L47: One large emission source that remains unmentioned here but is brought in later are FF emissions from often unregulated agricultural heavy machinery (L96), it could be mentioned here as well. We agree with the referee's comment. We mention this source in the Introduction section.

L54: This statement seems to be a gross exaggeration, and is also not supported by the reference provided. Biomass burning as a whole is indeed one of the largest sources of particles by mass worldwide, but the estimates for the contribution of agricultural fires are all over the place. This is to a large extent due to fact that non-wildfires BB sources are much less studied, and hence the current work is a welcome addition to the literature, but this needs to be qualified by a lot or rephrased. We have changed the structure of the introduction and removed the focus on agricultural burning. We removed the paragraph containing this sentence.

L60-62: If I follow, there is an important distinction to be made here: most AG burning is to clean out fields post-harvest. Sugarcane is burned as part of an expedited harvesting practice (although it might also be used post-harvest - this is not clear from the manuscript). This matters ultimately in terms of possible mitigation practices, so would suggest clarifying. We have clarified this with the following sentences: "The operations of sugarcane farming and harvesting, as well as the transport of the biomass to the mill factories, are all part of the sugar mill industry. Besides, the industrial process includes the use of sugarcane bagasse to cogenerate energy in boilers."

L67: Not sure how that reference is relevant. Also, you already mentioned AG burning field studies in L57-59, and those were not out of Europe/North America. In this paragraph, we focus on ambient PM chemical composition and it use to infer the impact of different sources, rather than agricultural burning, so we find the reference relevant.

L84: I think "center" makes more sense here than "centroid". Change was made.

L102: The source for the non-sugarcane emission inventories should be mentioned here as well, not just in the table.

L117: I think you mean "attribution", not "identification" (which has already happened). Change was made

L154: Source for the fire events? The source was the Regional Environmental Authority, which receives it from sugar companies. Change was made in the manuscript

L212: "non-combustion" Change was made.

L218: same comment as before Change was made.

L507: "zero", not "cero" Change was made.

Figure 2: Please specify what the red points in the right side figures are. I assume they are the upper 10% outliers, but that's not explained. Also, I would strongly suggest to show the average direction as a function of time of the day in the right side plots as well, since the afternoon switch in wind direction is key to interpret the results.

Figure S1: Pleas use English abbreviations in the X-Axis. Also, indicate the source of this information

Figure S2: Same comment about the X-Axis. Also please consider making the legends larger, it's extremely hard to read atm. Also please add the source of these measurements

Figure S4: Legend for line+symbol trace missing

References

Andreae, M. O.: Emission of trace gases and aerosols from biomass burning – an updated assessment, Atmos. Chem. Phys., 19, 8523–8546, 2019.

Brock, C. A., Wagner, N. L., Anderson, B. E., Attwood, A. R., Beyersdorf, A., Campuzano-Jost, P., Carlton, A. G., Day, D. A., Diskin, G. S., Gordon, T. D., Jimenez, J. L., Lack, D. A., Liao, J., Markovic, M. Z., Middlebrook, A. M., Ng, N. L., Perring, A. E., Richardson, M. S., Schwarz, J. P., Washenfelder, R. A., Welti, A., Xu, L., Ziemba, L. D., and Murphy, D. M.: Aerosol optical properties in the southeastern United States in summer – Part 1: Hygroscopic growth, Atmos. Chem. Phys., 16, 4987–5007, 2016.

Guo, H., Xu, L., Bougiatioti, A., Cerully, K. M., Capps, S. L., Hite, J. R., Carlton, A. G., Lee, S.-H., Bergin, M. H., Ng, N. L., Nenes, A., and Weber, R. J.: Fine-particle water and pH in the southeastern United States, Atmos. Chem. Phys., 15, 5211–5228, 2015.

Heim, E. W., Dibb, J., Scheuer, E., Jost, P. C., Nault, B. A., Jimenez, J. L., Peterson, D., Knote, C., Fenn, M., Hair, J., Beyersdorf, A. J., Corr, C., and Anderson, B. E.: Asian dust observed during KORUS-AQ facilitates the uptake and incorporation of soluble pollutants during transport to South Korea, Atmos. Environ., 224, 117305, 2020.

Hodshire, A. L., Akherati, A., Alvarado, M. J., Brown-Steiner, B., Jathar, S. H., Jimenez, J. L., Kreidenweis, S. M., Lonsdale, C. R., Onasch, T. B., Ortega, A. M., and Pierce, J. R.:

Aging Effects on Biomass Burning Aerosol Mass and Composition: A Critical Review of Field and Laboratory Studies, Environ. Sci. Technol., 53, 10007–10022, 2019.

Jolleys, M. D., Coe, H., Mc Figgans, G., Taylor, J. W., O'Shea, S. J., Le Breton, M., -B. Bauguitte, S. J., Moller, S., Di Carlo, P., Aruffo, E., Palmer, P. I., Lee, J. D., Percival, C. J., and Gallagher, M. W.: Properties and evolution of biomass burning organic aerosol from Canadian boreal forest fires, Atmos. Chem. Phys., 15, 3077–3095, 2015.

Nault, B. A., Campuzano-Jost, P., Day, D. A., Jo, D. S., Schroder, J. C., Allen, H. M., Bahreini, R., Bian, H., Blake, D. R., Chin, M., Clegg, S. L., Colarco, P. R., Crounse, J. D., Cubison, M. J., DeCarlo, P. F., Dibb, J. E., Diskin, G. S., Hodzic, A., Hu, W., Katich, J. M., Kim, M. J., Kodros, J. K., Kupc, A., Lopez-Hilfiker, F. D., Marais, E. A., Middlebrook, A. M., Andrew Neuman, J., Nowak, J. B., Palm, B. B., Paulot, F., Pierce, J. R., Schill, G. P., Scheuer, E., Thornton, J. A., Tsigaridis, K., Wennberg, P. O., Williamson, C. J., and Jimenez, J. L.: Chemical transport models often underestimate inorganic aerosol acidity in remote regions of the atmosphere, Communications Earth & Environment, 2, 1–13, 2021.

Petters, M. D. and Kreidenweis, S. M.: A single parameter representation of hygroscopic growth and cloud condensation nucleus activity, Atmos. Chem. Phys., 7, 1961–1971, 2007.

Sullivan, A. P., Guo, H., Schroder, J. C., CampuzanoâJost, P., Jimenez, J. L., Campos, T., Shah, V., Jaeglé, L., Lee, B. H., LopezâHilfiker, F. D., Thornton, J. A., Brown, S. S., and Weber, R. J.: Biomass burning markers and residential burning in the WINTER aircraft campaign, J. Geophys. Res., 124, 1846–1861, 2019.

---

## Referee Report (RR1)

**2nd review of "Understanding aerosol composition in a tropical inter-Andean valley impacted by agro-industrial and urban emissions" by Mateus Fontecha et al**

The revised manuscript has certainly improved both in readability and discussion of some of the results. Given the limited dataset being discussed, the interpretation and context provided for the results seem sufficient to warrant publication in ACP, taking into account the lack of overall aerosol chemistry information for this part of the world. However, as already mentioned in the previous report, this should be accepted as a "Measurement Report", since given the (understandably) limited conclusions.

I am responding here mostly to the initial response by the authors (which was not complete), and have added some additional comments at the end. My original comments are in blue, responses by the authors in green, and my new responses in orange.

Overall I think the authors still need to streamline the acidity discussion a bit more. I would also strongly suggest shortening the MSA discussion (or moving part of it to SI), since it does not seem to add much our understanding of the CRV chemistry.

Response to Anonymous Referee #2

Major comments

Comment 1

While the data and methods are presented quite well, the manuscript seems unfocused in the introduction. As the abstract makes pretty clear, this is first and foremost an observational report from a rural/suburban area with some AG burning thrown in. So I would expect a discussion of similar studies both in South America as well as other rural areas. Instead, the intro is focused on sugarcane burning, which might be one motivation for this study but is certainly not its main goal nor result. So a reorganization of the intro would I think make clearer what this effort is about and also help shorten the overall length.

Authors Response:

The introduction was reorganized and the focus on sugarcane burning was changed. The introduction now includes information about the diversity of PM2.5 sources existing in CRV and the main studies on PM2.5 chemical characterization conducted in the region. The literature review shows that studies with a detailed chemical composition on PM2.5 in Latin America are scarce. Most of the studies found have been focused on main cities and a few other areas, but there is an important gap in knowledge on PM2.5 composition and sources agro-industrial areas.

I appreciate the effort the authors put into the reorganization of the manuscript. As noted above, it certainly reads better now in my opinion. However, there are some new typos that I list at the end of the review. Also, as discussed in the following responses, the discussion could be streamlined further in some places.

Comment 2

The paper clearly shows that there is plenty of primary aerosol in the CRV, both from fossil fuel (FF) combustion and agricultural burning (BB). However, the EC/OC ratios for these two types of sources are quite different (also b/c of some fraction of brown carbon getting assigned to EC in the sunset

instrument). So using a simple regression to differentiate primary and secondary OC in this case is likely not very accurate, as the authors concede. A multivariate analysis is often superior in separating the different contributions, but the PCA analysis shown later does not really seem to be providing a meaningful distinction. So assuming a better multivariate analysis is not available, it might be better to try to predict primary OC based on a simple linear model that uses e.g. levoglucosan or K+ (which as a non-volatile tracer is likely a better choice) as a predictor for primary BBOM, and the sum of the lower alkanes as a predictor for primary OM from FF. As a boundary condition, we know that for most types of primary BBOA, levoglucosan is 3-6% of the total OA, so 5-8% of OM (see e.g. Jolleys et al 2015 and Sullivan et al 2019). This approach will likely underestimate secondary OM, but it will possibly give us more insight on what conditions actually lead to these primary components being maximized.

Authors Response:

Attending the suggestion of the reviewer, a simple linear model was applied to find the proportion of OCprim from three sources that are significant in the CRV, namely fossil fuel combustion (OCFF), biomass burning (OCBB), and vegetable detritus (OCdet). OCFF, OCBB and OCdet was quantified using a linear model from the following tracers: BghiP and IcdP for fossil fuel; levoglucosan for biomass burning; and the sum of the highest molecular weight alkanes (C27 – C33) for vegetable detritus. As a result, we estimated that 16.4% of OC can be attributed to fossil fuel, 15.2% to biomass burning, and 1.5% to vegetable detritus. In total, 32.7% of OC can be attributed to primary OC from these sources. This result is consistent with the apportionment of OCprim obtained by the OC/EC method, which indicated that around 50% of OC can be attributed to primary emissions. This approach does not allow us to infer OCsec because we don't have information about specific organic tracers of this carbonaceous fraction, such as α-pinene, β-caryophyllene, naphthalene-derived, and isoprene-derived SOA. Therefore, we decided to use both methods in the manuscript, to estimate both OCprim and OCsec, and check the consistency of OCprim results.

I appreciate the inclusion of this analysis, but I am completely missing the detail of how you come up with these numbers. Are these fitted values on minimizing r2? Or are you (as I was suggesting) taking literature values for the typical fraction of your markers (if so which ones?) and scaling them? Both approaches can work, I think the second one is likely more robust given your limited dataset, but either way this needs to be documented better. It is very clear how you did the EC/OC analysis, but this part is not.

Comment 3

The values currently used for the OM to OC conversion are reasonable, and clearly contribute to a decent mass closure (which I think should be better documented by e.g. showing a timeseries of the mass closure for each filter). However, I am not following the reasoning behind some of the values discussed. For primary BB emissions, in his recent review, Andreae (2019) used 1.6, which is a number that is roughly consistent with the results from other recent primary BB studies (e.g. Hodshire et al, 2019). BBOA can oxidize quite quickly, so secondary BBOA can indeed exhibit much higher OA:OC ratios, but that's not relevant in that context. Likewise, levoglucosan is an excellent tracer for primary BB, and hence does not belong into a discussion of secondary OM/OC. Importantly, references for the 2.1 value used for secondary OM are completely missing.

Authors Response:

We agree with the reviewer that levoglucosan does not belong into a discussion of OCsec and its conversion to OMsec. Accordingly, we have removed the corresponding sentence. We have rewritten the paragraph to clarify the OM/OC ratios used and their references. Now it reads: "Turpin and Lim (2001a) recommended an OM/OC ratio of 1.6 ± 0.2 for urban aerosols, and 2.1 ± 0.2 for non-urban aerosols, values comparable with those found by Aiken et al. (2008), of 1.71 (1.41 – 2.15), where lower values (1.6 – 1.8) are attributed to ground measurements in the morning, and higher values (1.8 – 1.9) to aircraft sample measurements. BB aerosols can have even higher f values (2.2-2.6), due to the presence of organic components with higher molecular weights, e.g., levoglucosan. However, Andreae (2019) recommends a factor of 1.6 for fresh BB aerosol, which is consistent with Hodshire et al (2019). We believe that traffic and biomass burning are the dominant OCprim sources at our site. Therefore, we used f1 = 1.6 to estimate OMpri. We used a factor of 2.1 to estimate OMsec from the OCsec fraction. This factor was chosen based on i) recommended ratios of 2.1±0.2 for aged aerosols."

This is a good discussion. It might be worth also mentioning that 2.1 is also the standard OM/OC ratio in many newer global models (e.g. Tsigaridis et al, 2014, Pai et al, 2020)

Comment 4

I am puzzled by the discussion of particle acidity. It is obviously always problematic to infer acidity without taking into account the gas phase, but having said that, a Cation:Anion ratio of ~0.8 is in most cases indicative of a fairly acidic aerosol (e.g. not that different from the SE US, where pH~1 was reported by Guo et al, 2015). And as the authors write, the NH4:SO4 ratio suggest that the fine aerosol might be even more acidic (Cation:Anion ratio of ~0.6) and that sea salt and dust are probably biasing the bulk ratio high. This has a few important implications that I think should be discussed at greater length.

Authors Response:

We decided to estimate pH for each sample from the concentrations of ions NH4+, NO3-, SO42-, Cl- and Na+ using the E-AIM thermodynamic model. The estimated pH~2.5 indicates that the aerosol are slightly acid as was previously suggested from the [NH4+]/[SO42-] ratio, as well as cations/anions equivalent ratios. However, these ratios and pH were not well correlated (r2 = ), which would indicate that those ratios are not a robust indicator of aerosol acidity, as mentioned by (Pye et al., 2020). The [NH4+]/[SO42-] ratio observed in the CRV was similar to that reported by Guo et al., (2015) in Southeastern US. However, the pH in the CRV aerosol was significantly higher (pH = 2-4) than that in Southeastern US (pH < 1) (Pye et al., 2020). The pH in the CRV aerosol can be explained by the presence of sulfate and organic compounds (pH = 1-3), as well as some influence of BB aerosol (pH = 4-5). This was added to the manuscript in section 3.3.

I am very confused. In the manuscript, you write that:

"Despite the ions molar ratio do not is centered fact in the pH central concept, the correlation observed between [NH4 + ]/[SO4 2- ] and pH was strong (r2 = 0.96)"

The first part of the sentence makes no sense and has to be revised, but if I understand correctly this says the opposite of what the authors write above. Furthermore, this is not particularly surprising: In the

absence of gas inputs the only difference between the two is ultimately the water content, which mostly affects the activity, not the concentration of [H+]. According to Pye et al, log([H+]) (ph_f in that paper) should be used for pH, not the activity coefficient, as the authors write in Table 1. So while this should probably corrected, it clealy has no impact on the results. So the pH is basically just reproducing the info put into the model, e.g. the cation balance. While not ideal, I think taken together your analysis makes a strong case that the real pH is indeed acidic and probably around ~2 or so.

Comment 5

If the aerosol is this acidic. nitrate is mostly going to partition to the gas phase, which is essentially what the data shows (although as mentioned below, some volatilization might play a role here). So nitrate from both NOx emissions from primary combustion processes

is not going to be detected, and therefore there is no point in interpreting NO3/SO4 ratios as indicative of any particular source. Furthermore, as discussed later in the PCA section, a significant fraction of nitrate seems to be associated with dust, so it is likely that there is no secondary NH4NO3 in this dataset at all. If so, the mass closure might improve by assuming all nitrate to be Ca(NO3)2 and not NH4NO3.

Authors' Response:

We added a discussion about the partitioning of total nitrate (NO3- + HNO3) and total ammonium (NH4+ + NH3) based on the acidity of aerosols in section 3.3. In this case, the aerosol acidity indicates that nitrate volatilization as HNO3 to the gas phase is favored, which is consistent with the lower concentration of NO3- (compere with SO42-). The [NH4+]/[SO42-] ratio (~1.6) indicates an environment with insufficient concentrations of ammonium to neutralize SO42- completely. Instead, sulfates seem to be partially neutralized as letovicite ([NH4]3H[SO4]2), as suggested by (Lee et al., 2008). Due to the estimated ammonium deficit, we suggest that nitrate salts were present in PM2.5 samples as KNO3, NaNO3, or Ca(NO3)2.

The use of [SO42-]/[NO3-] ratio to infer the origin of PM2.5 was removed.

This is an important result that actually supports the notion of a strongly acidic pH and would strongly suggest presenting it that way. If there is really no reason to assume the presence of NH4NO3, a pH of 2 or lower for the submicron mode is fairly likely.

Comment 6

But more importantly, given that both agriculture and BB are large sources of NH3(g), this is a very surprising result and I would encourage the authors to try to understand how it comes about. Given the overall composition, a fairly large source of SO2/H2SO4 would be needed to sustain such an imbalance (e.g. like the large SO2 emissions from coal fired plants in the SE US). But that does not seem to be the case here. Alternatively, the marine aerosol coming off the Walker Circulation might be quite acidic (e.g Nault et al, 2021), but that would imply that most of the sulfate is of marine origin, which seems unlikely.

Authors Response:

In order to explain the high concentrations of SO42- observed in PM2.5, we added a description of the diversity of sources of sulfur-containing compounds (SO2, SO3 and H2S) in the CRV, including coal in

power plants and industrial facilities, the biomass burning activities, the emission of H2S in poultry production animal production system, and the consumption of diesel for vehicles and tractors used to collect the sugarcane harvest (see section 3.3). The high concentration of SO42- and its low correlation with typical soluble ions associated with sea salt aerosol, suggests that Walker Circulation is not the main source of sulfate.

The absence of seasalt is certainly a good argument in terms of ruling out strong marine aerosol influences. So it seems that this is again a case like the SE US or Western China where coal burning produces enough sulfuric acid to pretty much overwhelm everything else and hence explains the inorganic aerosol distributions. BUT in L548 the authors write that coal combustion is fairly limited in the CRV, and the AG H2S sources you mention don't seem strong enough to compensate for the ammonia emissions. So what are we missing here? Is it possible that diesel combustion is still a major SO2 source? Otherwise it seems that the source of the acidity is really missing.

In terms of the ultimate conclusion, I think a more direct way to summarize this is to say that for the CRV the ratio of accumulation to coarse mode is fairly high (e.g both dust+seasalt, which are the main sources of alkalinity in most environments, are <5%) and hence the bulk acidity trends low. I think mentioning sulfate muddles this, since there is also sulfate present in seasalt and some dusts, so I would suggest focusing both here and in the abstract on the predominance of accumulation mode aerosol.

**Comment 7**

Related to the previous point, I wonder if there are published MSA/SO4 (or SO4/seasalt) ratios for the coastal areas in Colombia that could be used to address one of the larger open questions in this study, namely how much background aerosol from the ocean (and also the mountains when the katabatic winds kick in) contributes to the air quality in the CRV.

Authors Response:

PM2.5 chemical composition studies in the region are scarce and none of this has reported the concentration of methanesulfonate until now. We used the [methanesulfonate]/[SO42-] ratio to infer the impact of biogenic sulfur compounds on the total SO4-2 levels. 23.6 ± 20.1 showing the abundance or between Na+ and was

This response seems to be incomplete.

More generally, regarding the MSA discussion: as indicated by the authors, there are (not-well understood) anthropogenic sources of MSA, so calling it a marker for biogenic sulfate without qualification is not quite right, suggest revising. Ultimately it seems that the time resolution of the dataset is just not high enough to answer this specific question well, and this would certainly be a prime target for followup studies with 6-12 h time resolution or better.

**Comment 8**

In its current form, the PCA analysis does not really contribute much to our understanding of the data, especially since the factors are not really discussed in the context of the rich marker analysis introduced before. But regardless how it is presented, while the factors for FF combustion make some intuitive sense, that's not the case for the others. E.g. having all the secondary and marine aerosols in one factor (factor 2) probably mostly reflects that all of these come from the same wind direction and have no

strong correlations with other factor. Likewise, I was surprised to see levoglucosan in the road dust factor and Cl in one of the dust factors, but not associated with MSA/Na. To be clear, multivariate analysis is often messy, so some oddities are normal, but overall this analysis seems unfinished and tacked on. As the other reviewer already mentioned, this could at least partially be due to PCA not being the best tool to analyze this type of data, especially given how many low S/N tracers are included in the input matrix. But regardless of the choice of method, the whole point of such an analysis is to simplify the trends in the data, and it clearly fails at that. So I would suggest either eliminating it completely, and instead spend more time discussing how to combine the different findings from the marker analysis (which is currently missing), or try to find a new solution, either with PCA or other techniques (e.g. PMF) that is more suitable. In the end, the dataset might just be too flat to get good factors out of it, so the marker discussion approach seems more promising.

Authors Response:

Following the suggestion of Referee #2, we removed the PCA section from the manuscript. In fact, the preliminary PCA solution did not allow us to draw useful conclusions regarding the significance of sources in the CRV. We recognize that PCA is an ordinary least square fit of parametric variables, that does not include the uncertainty of each measurement and its variability, as indicated by Hopke, (2015). Therefore, its applicability is very limited in comparison with more robust techniques such as PMF or CBM. We will address source contribution analysis in a future study, so we focused on chemical species concentrations and diagnostic ratios in this manuscript.

Ok

Minor/technical comments:

All the abstract level numbers seem to be at least 1-2 digits over their actual significance, consider revising. The number of digits was adjusted

Ok

All Figures (except maybe Figure 3): please increase the font of the axis by at least 30-50%, very hard to read. The font size in figures was increased

Ok

A dust fraction of 9% in PM2.5 is quite high, and would suggest that PM10 is quite a bit larger than PM2.5, this should probably be mentioned in the discussion of inventories vs measurements. The re-calculated result for this fraction, with the adjustments made, was 3.5%, which is less than half of the previous value. Regarding emission inventories,

Answer seems incomplete. Ok on dust.

Semivolatile components of PM, such as ammonium nitrate and SVOA can easily evaporate from filters, biasing the fractional and overall composition measurement (see e.g. Heim et al, 2020 for a recent example). The sampling site seems to have been fair enough removed from primary sources that this is probably a minor concern, but it is a common filter artifact that should be mentioned.

This has not been addressed either here nor in the manuscript

Given how stable the RH was during sampling (which should probably be better emphasized when discussing this), there is nothing wrong with estimating a constant particulate water contribution with E-AIM, although it might be simpler and more comparable to just use speciated kappa values instead (e.g. Peters et al 2007, Brock et al 2016).

This has not been addressed either here nor in the manuscript

As a side note, I am a bit surprised that the authors used E-AIM Model II for the water estimation on Model IV for pH… Should not a make a difference, but for consistency it would probably be best to just pick one?

I am not too familiar with the CPI measure, but I wonder if BB emission could lead to e.g. a bias towards non-FF emissions, please discuss. The CPI for fossil fuel combustion emissions is around 1, whereas for sugarcane burning is around 2. The occurrence of BB will produce a value higher than 1 for CPI in ambient aerosol, and it would be close to 2 for areas heavily impacted by BB. In this study, a value of 1.2 suggests a moderate influence of BB and vegetative detritus in ambient aerosol.

Thanks for the clarification.

L44-L47: One large emission source that remains unmentioned here but is brought in later are FF emissions from often unregulated agricultural heavy machinery (L96), it could be mentioned here as well. We agree with the referee's comment. We mention this source in the Introduction section.

Ok. Going back to the sulfur in diesel comment above, I wonder if specifically the fuel for these machines is our missing source?

L54: This statement seems to be a gross exaggeration, and is also not supported by the reference provided. Biomass burning as a whole is indeed one of the largest sources of particles by mass worldwide, but the estimates for the contribution of agricultural fires are all over the place. This is to a large extent due to fact that non-wildfires BB sources are much less studied, and hence the current work is a welcome addition to the literature, but this needs to be qualified by a lot or rephrased. We have changed the structure of the introduction and removed the focus on agricultural burning. We removed the paragraph containing this sentence.

Ok

L60-62: If I follow, there is an important distinction to be made here: most AG burning is to clean out fields post-harvest. Sugarcane is burned as part of an expedited harvesting practice (although it might also be used post-harvest - this is not clear from the manuscript). This matters ultimately in terms of possible mitigation practices, so would suggest clarifying. We have clarified this with the following sentences: "The operations of sugarcane farming and harvesting, as well as the transport of the biomass to the mill factories, are all part of the sugar mill industry. Besides, the industrial process includes the use of sugarcane bagasse to cogenerate energy in boilers."

Ok

L67: Not sure how that reference is relevant. Also, you already mentioned AG burning field studies in L57-59, and those were not out of Europe/North America. In this paragraph, we focus on ambient PM

chemical composition and it use to infer the impact of different sources, rather than agricultural burning, so we find the reference relevant.

Thanks for the clarification.

L84: I think "center" makes more sense here than "centroid". Change was made.

Ok

L102: The source for the non-sugarcane emission inventories should be mentioned here as well, not just in the table.

Response missing, but I think the new intro addresses this.

L117: I think you mean "attribution", not "identification" (which has already happened). Change was made

Ok

L154: Source for the fire events? The source was the Regional Environmental Authority, which receives it from sugar companies. Change was made in the manuscript

Thanks. So this could well be a lower limit due to underreporting. Up to the authors if they want to emphasize this.

L212: "non-combustion" Change was made.

Ok

L218: same comment as before Change was made.

Ok

L507: "zero", not "cero" Change was made.

Ok

Figure 2: Please specify what the red points in the right side figures are. I assume they are the upper 10% outliers, but that's not explained. Also, I would strongly suggest to show the average direction as a function of time of the day in the right side plots as well, since the afternoon switch in wind direction is key to interpret the results.

This comment has not been addressed either here or in the revised manuscript

Figure S1: Pleas use English abbreviations in the X-Axis. Also, indicate the source of this information

This comment has not been addressed either here or in the revised manuscript.

Figure S2: Same comment about the X-Axis. Also please consider making the legends larger, it's extremely hard to read atm. Also please add the source of these measurements

This comment has not been addressed either here or in the revised manuscript

Figure S4: Legend for line+symbol trace missing

This comment has not been addressed either here or in the revised manuscript

**New comments:**

L318: While hard to estimate, it might be worth pointing out that the wind distribution does suggest that near-surface concentrations (which are most relevant for health outcomes) might be substantially higher than presented here, especially in the summer months where stagnation can increase production of secondary species. It is also unclear to the reviewer to what extent this bias could potentially impact the comparisons with the other sugarcane studies mentioned in the next paragraph. Were those done closer to the ground?

Most of the Figures could be revised for better legibility and clarity. In particular:

Figure 1: Resolution seems a bit degreaded, would try to ensure that its higher res for publication, some symbols are hard to see

Figure 3: A smaller axis range for levoglucosan would show the differences more clearly. Also, why not show the filter data from Fig 5 (as solid bars, not speciated) instead of PM2.5, given the much better spatial coverage and better correlation overall? I understand this is not specifically what Reviewer #1 asked for, but it would seem like the more appropriate Figure (and could go into SI as well if after replotting it does not really show any clear trends, which is likely).

Figure 4 would suggest adding a legend on the right of the two figures with the full names of the individual PAHs, that would make digesting it easier

Figure S6 is obsolete for the revised manuscript.

Minor comments

L30: Incomplete sentence, please revise

L33: "PAHs" is I think the more common usage

L42: "of" not needed

L70: If providing the exact number of vehicles, either a cite or at a minimum a year of reporting should be mentioned

L76: I take it that there is no widespread use of small aircraft (which are typically very polluting) for insecticide application and other farming activities, correct? Otherwise please elaborate

L81 " and an estimation" (space missing)

L85: I assume you mean that an estimate of total emissions is not possible with these data, unlikely individual sources. Maybe worth clarifying?

L103: "air quality due to the agroindustry" is what you mean?

L105: "industrial sources and fertilizer use" would read better

L113-118: Probably nothing wrong with pointing out here that this study is meant to motivate future research activities that will allow for full source analysis

L237-L241: Has this been done before? References?

L358: Garbled sentence, do not follow

L361: This is mostly a summary of Pye's findings, please note it as such.

L369: "remaining as NH4+ IN the aerosol" is I believe what the authors mean here

L372: "in the absence of submicron seasalt and dust" is an important caveat here (as discussed later in L394). As noted above, it would seem to make sense to restructure the discussion and show first that nitrate is mostly not associated with NH4 and then introduce this as a supporting fact for the acidity discussion.

---

## Author Response (AR2)

Manuscript: Understanding aerosol composition in an inter-Andean valley impacted by sugarcane intensive agriculture and urban emissions

Authors: Lady Mateus-Fontecha, Angela Vargas-Burbano, Rodrigo Jimenez, Nestor Y. Rojas, German Rueda-Saa, Dominik van Pinxteren, Manuela van Pinxteren, Khanneh Wadinga Fomba, and Hartmut Herrmann

Overall, I think the authors still need to streamline the acidity discussion a bit more. I would also strongly suggest shortening the MSA discussion (or moving part of it to SI), since it does not seem to add much our understanding of the CRV chemistry.

We appreciate all comments and contributions made by the reviewer. Following we answer each comment and observation made by editor in chief.

Figure 1, which contains a map was created by us. The acknowledgments section includes credit to member of research group who create the map.

We changed the paragraph where the methanesulphonate concentrations and relationships with other soluble ions are presented (lines 429 - 430). Given the high correlation between methanesulphonate and sulphate and the non-negligible correlation between methanesulphonate and oxalate, we think it is valuable and useful to suggest future studies to not only associate the presence of methanesulphonate with the oxidation of DMS produced in seawater.

Major comments

Comment 1
While the data and methods are presented quite well, the manuscript seems unfocused in the introduction. As the abstract makes pretty clear, this is first and foremost an observational report from a rural/suburban area with some AG burning thrown in. So I would expect a discussion of similar studies both in South America as well as other rural areas. Instead, the intro is focused on sugarcane burning, which might be one motivation for this study but is certainly not its main goal nor result. So a reorganization of the intro would I think make clearer what this effort is about and also help shorten the overall length.

Authors Response:
The introduction was reorganized and the focus on sugarcane burning was changed. The introduction now includes information about the diversity of PM2.5 sources existing in CRV and the main studies on PM2.5 chemical characterization conducted in the region. The literature review shows that studies with a detailed chemical composition on PM2.5 in Latin America are scarce. Most of the studies found have been focused on main cities and a few other areas, but there is an important gap in knowledge on PM2.5 composition and sources agro-industrial areas.

I appreciate the effort the authors put into the reorganization of the manuscript. As noted above, it certainly, reads better now in my opinion. However, there are some new typos that I list at the end of the review. Also, as discussed in the following responses, the discussion could be streamlined further in some places.

We made improvements in response to the feedback, and the discussion was streamlined, particularly in MSA and the attribution of OC primary.

Comment 2
The paper clearly shows that there is plenty of primary aerosol in the CRV, both from fossil fuel (FF)

combustion and agricultural burning (BB). However, the EC/OC ratios for these two types of sources are quite different (also b/c of some fraction of brown carbon getting assigned to EC in the sunset instrument). So using a simple regression to differentiate primary and secondary OC in this case is likely not very accurate, as the authors concede. A multivariate analysis is often superior in separating the different contributions, but the PCA analysis shown later does not really seem to be providing a meaningful distinction. So assuming a better multivariate analysis is not available, it might be better to try to predict primary OC based on a simple linear model that uses e.g. levoglucosan or K+ (which as a non-volatile tracer is likely a better choice) as a predictor for primary BBOM, and the sum of the lower alkanes as a predictor for primary OM from FF. As a boundary condition, we know that for most types of primary BBOA, levoglucosan is 3-6% of the total OA, so 5-8% of OM (see e.g. Jolleys et al 2015 and Sullivan et al 2019). This approach will likely underestimate secondary OM, but it will possibly give us more insight on what conditions actually lead to these primary components being maximized.

Authors Response:
Attending the suggestion of the reviewer, a simple linear model was applied to find the proportion of OCprim from three sources that are significant in the CRV, namely fossil fuel combustion (OCFF), biomass burning (OCBB), and vegetable detritus (OCdet). OCFF, OCBB and OCdet was quantified using a linear model from the following tracers: BghiP and IcdP for fossil fuel; levoglucosan for biomass burning; and the sum of the highest molecular weight alkanes (C27 – C33) for vegetable detritus. As a result, we estimated that 16.4% of OC can be attributed to fossil fuel, 15.2% to biomass burning, and 1.5% to vegetable detritus. In total, 32.7% of OC can be attributed to primary OC from these sources. This result is consistent with the apportionment of OCprim obtained by the OC/EC method, which indicated that around 50% of OC can be attributed to primary emissions. This approach does not allow us to infer OCsec because we don't have information about specific organic tracers of this carbonaceous fraction, such as α-pinene, β-caryophyllene, naphthalene-derived, and isoprene-derived SOA. Therefore, we decided to use both methods in the manuscript, to estimate both OCprim and OCsec, and check the consistency of OCprim results.

I appreciate the inclusion of this analysis, but I am completely missing the detail of how you come up with these numbers. Are these fitted values on minimizing r2? Or are you (as I was suggesting) taking literature values for the typical fraction of your markers (if so which ones?) and scaling them? Both approaches can work, I think the second one is likely more robust given your limited dataset, but either way this needs to be documented better. It is very clear how you did the EC/OC analysis, but this part is not.

We applied fitted linear model by robust regression with a M estimator by bisquare function to estimate fossil fuel combustion (OCFF), biomass burning (OCBB), and vegetable detritus (OCdet) coefficients. The description of linear model used is in the lines 246-256.

Comment 3
The values currently used for the OM to OC conversion are reasonable, and clearly contribute to a decent mass closure (which I think should be better documented by e.g. showing a timeseries of the mass closure for each filter). However, I am not following the reasoning behind some of the values discussed. For primary BB emissions, in his recent review, Andreae (2019) used 1.6, which is a number that is roughly consistent with the results from other recent primary BB studies (e.g. Hodshire et al, 2019). BBOA can oxidize quite quickly, so secondary BBOA can indeed exhibit much higher OA:OC ratios, but that's not relevant in that context. Likewise, levoglucosan is an excellent tracer for primary BB, and hence does not belong into a discussion of secondary OM/OC. Importantly, references for the 2.1 value used for secondary OM are completely missing.

Authors Response:

We agree with the reviewer that levoglucosan does not belong into a discussion of OCsec and its conversion to OMsec. Accordingly, we have removed the corresponding sentence. We have rewritten the paragraph to clarify the OM/OC ratios used and their references. Now it reads: "Turpin and Lim (2001a) recommended an OM/OC ratio of 1.6 ± 0.2 for urban aerosols, and 2.1 ± 0.2 for non-urban aerosols, values comparable with those found by Aiken et al. (2008), of 1.71 (1.41 – 2.15), where lower values (1.6 – 1.8) are attributed to ground measurements in the morning, and higher values (1.8 – 1.9) to aircraft sample measurements. BB aerosols can have even higher f values (2.2-2.6), due to the presence of organic components with higher molecular weights, e.g., levoglucosan. However, Andreae (2019) recommends a factor of 1.6 for fresh BB aerosol, which is consistent with Hodshire et al (2019). We believe that traffic and biomass burning are the dominant OCprim sources at our site. Therefore, we used f1 = 1.6 to estimate OMpri. We used a factor of 2.1 to estimate OMsec from the OCsec fraction. This factor was chosen based on i) recommended ratios of 2.1±0.2 for aged aerosols."

This is a good discussion. It might be worth also mentioning that 2.1 is also the standard OM/OC ratio in many newer global models (e.g. Tsigaridis et al, 2014, Pai et al, 2020).

The description of the OM/OC ratios used by some global models was included in lines: 262 – 266.

Comment 4

I am puzzled by the discussion of particle acidity. It is obviously always problematic to infer acidity without taking into account the gas phase, but having said that, a Cation:Anion ratio of ~0.8 is in most cases indicative of a fairly acidic aerosol (e.g. not that different from the SE US, where pH~1 was reported by Guo et al, 2015). And as the authors write, the NH4:SO4 ratio suggest that the fine aerosol might be even more acidic (Cation : Anion ratio of ~0.6) and that sea salt and dust are probably biasing the bulk ratio high. This has a few important implications that I think should be discussed at greater Length.

Authors Response:
We decided to estimate pH for each sample from the concentrations of ions $NH_4^+$, $NO_3^-$, $SO_4^{2-}$, $Cl^-$ and $Na^+$ using the E-AIM thermodynamic model. The estimated pH~2.5 indicates that the aerosol are slightly acid as was previously suggested from the $[NH_4^+]/[SO_4^{2-}]$ ratio, as well as cations/anions equivalent ratios. However, these ratios and pH were not well correlated (r2 = ), which would indicate that those ratios are not a robust indicator of aerosol acidity, as mentioned by (Pye et al., 2020). The $[NH_4^+]/[SO_4^{2-}]$ ratio observed in the CRV was similar to that reported by Guo et al., (2015) in Southeastern US. However, the pH in the CRV aerosol was significantly higher (pH = 2-4) than that in Southeastern US (pH < 1) (Pye et al., 2020). The pH in the CRV aerosol can be explained by the presence of sulfate and organic compounds (pH = 1-3), as well as some influence of BB aerosol (pH = 4-5). This was added to the manuscript in section 3.3.

I am very confused. In the manuscript, you write that:
"Despite the ion's molar ratio do not is centered fact in the pH central concept, the correlation observed between $[NH_4^+]/[SO_4^{2-}]$ and pH was strong (r2 = 0.96)"

The first part of the sentence makes no sense and has to be revised, but if I understand correctly this says the opposite of what the authors write above.

The first part of the sentence was changed in lines: 377 - 378.

Furthermore, this is not particularly surprising: In the absence of gas inputs the only difference between the two is ultimately the water content, which mostly affects the activity, not the concentration of [H+]. According to Pye et al, log([H+]) (ph_f in that paper) should be used for pH,

not the activity coefficient, as the authors write in Table 1. So while this should probably corrected, it clealy has no impact on the results. So the pH is basically just reproducing the info put into the model, e.g. the cation balance. While not ideal, I think taken together your analysis makes a strong case that the real pH is indeed acidic and probably around ~2 or so.

The Eq (9) was corrected changing the activity coefficients by (H+) concentration. All in Line 292.

Comment 5.

If the aerosol is this acidic. nitrate is mostly going to partition to the gas phase, which is essentially what the data shows (although as mentioned below, some volatilization might play a role here). So nitrate from both NOx emissions from primary combustion processes is not going to be detected, and therefore there is no point in interpreting NO3/SO4 ratios as indicative of any particular source. Furthermore, as discussed later in the PCA section, a significant fraction of nitrate seems to be associated with dust, so it is likely that there is no secondary NH4NO3 in this dataset at all. If so, the mass closure might improve by assuming all nitrate to be Ca(NO3)2 and not NH4NO3.

Authors' Response:

We added a discussion about the partitioning of total nitrate (NO3- + HNO3) and total ammonium (NH4+ + NH3) based on the acidity of aerosols in section 3.3. In this case, the aerosol acidity indicates that nitrate volatilization as HNO3 to the gas phase is favored, which is consistent with the lower concentration of NO3- (compere with SO42-). The [NH4+]/[SO42-] ratio (~1.6) indicates an environment with insufficient concentrations of ammonium to neutralize SO42- completely. Instead, sulfates seem to be partially neutralized as letovicite ([NH4]3H[SO4]2), as suggested by (Lee et al., 2008). Due to the estimated ammonium deficit, we suggest that nitrate salts were present in PM2.5 samples as KNO3, NaNO3, or Ca(NO3)2.

The use of [SO42-]/[NO3-] ratio to infer the origin of PM2.5 was removed.

This is an important result that actually supports the notion of a strongly acidic pH and would strongly suggest presenting it that way. If there is really no reason to assume the presence of NH4NO3, a pH of 2 or lower for the submicron mode is fairly likely.

Done in line 408 - 414

Comment 6
But more importantly, given that both agriculture and BB are large sources of NH3(g), this is a very surprising result and I would encourage the authors to try to understand how it comes about. Given the overall composition, a fairly large source of SO2/H2SO4 would be needed to sustain such an imbalance (e.g. like the large SO2 emissions from coal fired plants in the SE US). But that does not seem to be the case here. Alternatively, the marine aerosol coming off the Walker Circulation might be quite acidic (e.g Nault et al, 2021), but that would imply that most of the sulfate is of marine origin, which seems unlikely.

Authors Response:

In order to explain the high concentrations of SO42- observed in PM2.5, we added a description of the diversity of sources of sulfur-containing compounds (SO2, SO3 and H2S) in the CRV, including coal in power plants and industrial facilities, the biomass burning activities, the emission of H2S in poultry production animal production system, and the consumption of diesel for vehicles and tractors used to collect the sugarcane harvest (see section 3.3). The high concentration of SO42- and its low correlation

with typical soluble ions associated with sea salt aerosol, suggests that Walker Circulation is not the main source of sulfate.

The absence of seasalt is certainly a good argument in terms of ruling out strong marine aerosol influences. So it seems that this is again a case like the SE US or Western China where coal burning produces enough sulfuric acid to pretty much overwhelm everything else and hence explains the inorganic aerosol distributions. BUT in L548 the authors write that coal combustion is fairly limited in the CRV, and the AG H2S sources you mention don't seem strong enough to compensate for the ammonia emissions. So what are we missing here? Is it possible that diesel combustion is still a major SO2 source? Otherwise it seems that the source of the acidity is really missing.

In terms of the ultimate conclusion, I think a more direct way to summarize this is to say that for the CRV the ratio of accumulation to coarse mode is fairly high (e.g both dust+seasalt, which are the main sources of alkalinity in most environments, are <5%) and hence the bulk acidity trends low. I think mentioning sulfate muddles this, since there is also sulfate present in seasalt and some dusts, so I would suggest focusing both here and in the abstract on the predominance of accumulation mode aerosol.

A preliminary inventory of SO2 was explored in order to try to explained the high concentrations of SO42- found in PM2.5. This information was included in Table S1, among is highlight brick kiln and artisanal coal kiln.
To try to explain the high amounts of SO42- observed in PM2.5, a preliminary inventory of SO2 was added. The amount of showcase brick kiln and artisanal coal kiln was reported in Table S1.

Comment 7

Related to the previous point, I wonder if there are published MSA/SO4 (or SO4/seasalt) ratios for the coastal areas in Colombia that could be used to address one of the larger open questions in this study, namely how much background aerosol from the ocean (and also the mountains when the katabatic winds kick in) contributes to the air quality in the CRV.

Authors Response:

PM2.5 chemical composition studies in the region are scarce and none of this has reported the concentration of methanesulfonate until now. We used the [methanesulfonate]/[SO42-] ratio to infer the impact of biogenic sulfur compounds on the total SO4-2 levels. 23.6 ± 20.1 showing the abundance or between Na+ and was

This response seems to be incomplete.

More generally, regarding the MSA discussion: as indicated by the authors, there are (not-well understood) anthropogenic sources of MSA, so calling it a marker for biogenic sulfate without qualification is not quite right, suggest revising. Ultimately it seems that the time resolution of the dataset is just not high enough to answer this specific question well, and this would certainly be a prime target for followup studies with 6-12 h time resolution or better.

We modify the MSA discussion showing the most significant correlations with other compounds measured in the samples. Amount the components that have the most relevant correlations with methanesulphonate are $SO_4^{2-}$ , $NH_4^+$, oxalate ion, the metals Se and Fe, and the carbonaceous fraction OC and EC. Lines 429 - 430

The relationship between methasulfonate and common sea salt ions, such as Na+, does not provide enough evidence that this ion is produced by sea aerosols. While there is a stronger association with other molecules, this suggests that a combustion activity that emits sulphate could be a source of methansulphonate as well. According to Meinardi et al (2003), biomass burning can create dimethyl disulfide (DMDS), which can then oxidize in the environment to methane sulfonic acid (MSA), a precursor to methanesulfonate. To determine whether the source of methasulfonate is marine aerosols or other terrestrial activity, more investigations with a higher time resolution (6–12 hours) are needed to keep track of the variations.

Comment 8

In its current form, the PCA analysis does not really contribute much to our understanding of the data, especially since the factors are not really discussed in the context of the rich marker analysis introduced before. But regardless how it is presented, while the factors for FF combustion make some intuitive sense, that's not the case for the others. E.g. having all the secondary and marine aerosols in one factor (factor 2) probably mostly reflects that all of these come from the same wind direction and have no strong correlations with other factor. Likewise, I was surprised to see levoglucosan in the road dust factor and Cl in one of the dust factors, but not associated with MSA/Na. To be clear, multivariate analysis is often messy, so some oddities are normal, but overall this analysis seems unfinished and tacked on. As the other reviewer already mentioned, this could at least partially be due to PCA not being the best tool to analyze this type of data, especially given how many low S/N tracers are included in the input matrix. But regardless of the choice of method, the whole point of such an analysis is to simplify the trends in the data, and it clearly fails at that. So I would suggest either eliminating it completely, and instead spend more time discussing how to combine the different findings from the marker analysis (which is currently missing), or try to find a new solution, either with PCA or other techniques (e.g. PMF) that is more suitable. In the end, the dataset might just be too flat to get good factors out of it, so the marker discussion approach seems more promising.

Authors Response:
Following the suggestion of Referee #2, we removed the PCA section from the manuscript. In fact, the preliminary PCA solution did not allow us to draw useful conclusions regarding the significance of sources in the CRV. We recognize that PCA is an ordinary least square fit of parametric variables, that does not include the uncertainty of each measurement and its variability, as indicated by Hopke, (2015). Therefore, its applicability is very limited in comparison with more robust techniques such as PMF or CBM. We will address source contribution analysis in a future study, so we focused on chemical species concentrations and diagnostic ratios in this manuscript.

Ok

Minor/technical comments:

All the abstract level numbers seem to be at least 1-2 digits over their actual significance, consider revising. The number of digits was adjusted
Ok
All Figures (except maybe Figure 3): please increase the font of the axis by at least 30-50%, very hard to
read. The font size in figures was increased
Ok

A dust fraction of 9% in PM2.5 is quite high, and would suggest that PM10 is quite a bit larger than PM2.5, this should probably be mentioned in the discussion of inventories vs measurements.

The recalculated result for this fraction, with the adjustments made, was 3.5%, which is less than half of the previous value.

Answer seems incomplete. Ok on dust.

Semivolatile components of PM, such as ammonium nitrate and SVOA can easily evaporate from filters, biasing the fractional and overall composition measurement (see e.g. Heim et al, 2020 for a recent example). The sampling site seems to have been fair enough removed from primary sources that this is probably a minor concern, but it is a common filter artifact that should be mentioned.

This has not been addressed either here nor in the manuscript

Lines 172 - 178 describe the possible uncertainty factors associated with sampling handling interferences.

Given how stable the RH was during sampling (which should probably be better emphasized when discussing this), there is nothing wrong with estimating a constant particulate water contribution with EAIM, although it might be simpler and more comparable to just use speciated kappa values instead (e.g. Peters et al 2007, Brock et al 2016).

This has not been addressed either here nor in the manuscript

As a side note, I am a bit surprised that the authors used E-AIM Model II for the water estimation on Model IV for pH… Should not a make a difference, but for consistency it would probably be best to just pick one?

We used AIM Model IV for the water and pH estimation.

L44-L47: One large emission source that remains unmentioned here but is brought in later are FF emissions from often unregulated agricultural heavy machinery (L96), it could be mentioned here as well. We agree with the referee's comment. We mention this source in the Introduction section.

Ok. Going back to the sulfur in diesel comment above, I wonder if specifically the fuel for these machines is our missing source?

The content sulfur in Colombian Diesel is 0.0063%.

L154: Source for the fire events? The source was the Regional Environmental Authority, which receives it from sugar companies. Change was made in the manuscript.

Thanks. So this could well be a lower limit due to underreporting. Up to the authors if they want to emphasize this.

Figure 2: Please specify what the red points in the right side figures are. I assume they are the upper 10% outliers, but that's not explained. Also, I would strongly suggest to show the average direction as a function of time of the day in the right side plots as well, since the afternoon switch in wind direction is key to interpret the results.

This comment has not been addressed either here or in the revised manuscript

Done. The red points were explained in the caption and the wind rose was plotted separated by day and night.

Figure S1: Pleas use English abbreviations in the X-Axis. Also, indicate the source of this information

This comment has not been addressed either here or in the revised manuscript.
Done

Figure S2: Same comment about the X-Axis. Also please consider making the legends larger, it's extremely hard to read atm. Also please add the source of these measurements.

This comment has not been addressed either here or in the revised manuscript.

Done

Figure S4: Legend for line+symbol trace missing

New comments:

L318: While hard to estimate, it might be worth pointing out that the wind distribution does suggest that near-surface concentrations (which are most relevant for health outcomes) might be substantially higher than presented here, especially in the summer months where stagnation can increase production of secondary species. It is also unclear to the reviewer to what extent this bias could potentially impact the comparisons with the other sugarcane studies mentioned in the next paragraph. Were those done closer to the ground?

Most of the Figures could be revised for better legibility and clarity. In particular:

Figure 1: Resolution seems a bit degreaded, would try to ensure that its higher res for publication, some symbols are hard to see

Done. The Fig 1 was changed.

Figure 3: A smaller axis range for levoglucosan would show the differences more clearly. Also, why not show the filter data from Fig 5 (as solid bars, not speciated) instead of PM2.5, given the much better spatial coverage and better correlation overall? I understand this is not specifically what Reviewer #1 asked for, but it would seem like the more appropriate Figure (and could go into SI as well if after replotting it does not really show any clear trends, which is likely).

Done. The axis of levoglucosan was reduced, improving the resolution of this scale.  bar plot don't look well.

Figure 4 would suggest adding a legend on the right of the two figures with the full names of the individual PAHs, that would make digesting it easier

Done. The legend on the right were adding with full name the individual PAHs.

Figure S6 is obsolete for the revised manuscript.
The Fig S6 don't exist.

Minor comments
L30: Incomplete sentence, please revise. Corrected

L33: "PAHs" is I think the more common usage. Changed along the manuscript

L42: "of" not needed. Changed

L70: If providing the exact number of vehicles, either a cite or at a minimum a year of reporting should be mentioned.

The reference is the Colombian base data of vehicles.

L76: I take it that there is no widespread use of small aircraft (which are typically very polluting) for insecticide application and other farming activities, correct? Otherwise please elaborate

Done lines 75 – 77.

L81 " and an estimation" (space missing). Corrected

L85: I assume you mean that an estimate of total emissions is not possible with these data, unlikely individual sources. Maybe worth clarifying?. The text was slightly modified for clarity

L103: "air quality due to the agroindustry" is what you mean?. Changed

L105: "industrial sources and fertilizer use" would read better. Changed

L113-118: Probably nothing wrong with pointing out here that this study is meant to motivate future research activities that will allow for full source analysis. We added a sentence in this sense.

L237-L241: Has this been done before? References?

We use a linear model as was suggested in the first revision by the referee #2 in the comment #2. This is a linear model by robust regression. In this model, we use several tracers as BghiP, IcdP, levoglucosan and the sum of highest alkanes (C27-C33) to explain the fraction of OCprim come from fossil fuel, biomass burning and detritus. One of the differences with the methodology used by Sullivan (2019) is that they made a separate correlation made for each tracer, while we use aggregate all tracers in a linear model.

L358: Garbled sentence, do not follow. The paragraph was modified

L361: This is mostly a summary of Pye's findings, please note it as such. The paragraph was modified

L369: "remaining as NH4+ IN the aerosol" is I believe what the authors mean here Changed

L372: "in the absence of submicron seasalt and dust" is an important caveat here (as discussed later in L394). As noted above, it would seem to make sense to restructure the discussion and show first that nitrate is mostly not associated with NH4 and then introduce this as a supporting fact for the acidity discussion.